# Single nuclei transcriptomics in human and non-human primate striatum in opioid use disorder

BaDoi N. Phan [1,2,3], Madelyn H. Ray[4,5], Xiangning Xue [6], Chen Fu[7], Robert J. Fenster[8,9], Stephen J. Kohut [8,10], Jack Bergman[8,10], Suzanne N. Haber[8,11], Kenneth M. McCullough [12], Madeline K. Fish [13,14], Jill R. Glausier [15], Qiao Su[1], Allison E. Tipton[13,14], David A. Lewis [15], Zachary Freyberg [15,16], George C. Tseng [6], Shelley J. Russek[4,13,14], Yuriy Alekseyev [17], Kerry J. Ressler [8,9], Marianne L. Seney [15], Andreas R. Pfenning [1,2] ✉ & Ryan W. Logan [4,7,18] ✉

In brain, the striatum is a heterogenous region involved in reward and goal-directed behaviors. Striatal dysfunction is linked to psychiatric disorders, including opioid use disorder (OUD). Striatal subregions are divided based on neuroanatomy, each with unique roles in OUD. In OUD, the dorsal striatum is involved in altered reward processing, formation of habits, and development of negative affect during withdrawal. Using single nuclei RNA-sequencing, we identified both canonical (e.g., dopamine receptor subtype) and less abundant cell populations (e.g., interneurons) in human dorsal striatum. Pathways related to neurodegeneration, interferon response, and DNA damage were significantly enriched in striatal neurons of individuals with OUD. DNA damage markers were also elevated in striatal neurons of opioid-exposed rhesus macaques. Sex-specific molecular differences in glial cell subtypes associated with chronic stress were found in OUD, particularly female individuals. Together, we describe different cell types in human dorsal striatum and identify cell type-specific alterations in OUD.

Fatal opioid overdoses and people diagnosed with opioid use disorder (OUD) are continuing to rise in the United States[1]. Efforts to develop new treatment strategies and to bolster avenues of existing treatments for opioid addiction require a deeper understanding of the changes that occur in the human brain with chronic opioid use. To date, few studies have investigated the molecular changes associated with OUD at the cellular level in human brain[2–8]. Recently, we and others have reported alterations in inflammatory signaling[5,9], along with pathways involved in neurodegeneration[2,6,7], oxidative stress[10], and DNA damage[11] in the brains of individuals with OUD. Glial cells are likely major contributors to inflammation in the brain associated with chronic opioid use[5,9].

In the brain, inflammation is initiated by a myriad of processes. For example, accumulation of DNA damage and associated factors can lead to the release of proinflammatory cytokines via microglia activation[12,13]. Prolonged proinflammatory states in the brain result in elevated oxidative stress and altered metabolism in glia[12–15] and neurons[13,16,17]. Neuroinflammation and DNA damage are also central to neurodegenerative processes[2,5,9], which have been related to OUD[18–23]. With the advent of high-throughput single cell technologies, such as single nuclei RNA-sequencing (snRNA-seq), deeper investigations into the roles of inflammation and DNA damage in OUD can be achieved in human brain. Beyond resolving cell type-specific molecular alterations previously found with other approaches (e.g., bulk tissue RNA-seq)[5],

snRNA-seq in human brain also offers the possibility of unmasking new biological relationships between neural cell types and disease.

A brain region implicated in OUD, along with other psychiatric and neurological disorders, is the striatum. In the human brain, the striatum is anatomically divided into dorsal and ventral portions, with both the caudate and putamen recognized as subregions of the dorsal striatum. Functional changes in the caudate and putamen are involved in altered reward processing, habitual drug-seeking and relapse, along with the development of negative affect during opioid withdrawal[24–27], underlying the key roles of the dorsal striatum in OUD. However, only a few studies have examined specific cell types in human dorsal striatum[28], and to date, no study has resolved the cell type-specific molecular alterations in human striatum in OUD.

To begin to investigate OUD-associated alterations across striatal cell types, we conducted snRNA-seq on the dorsal striatum from human postmortem brain and compared the cell type-specific molecular signatures between unaffected individuals and individuals with OUD (98,848 total nuclei from the caudate and putamen of males and females, 12 individuals and 24 biological samples). Given the high number of high-quality nuclei per sample, we were able to identify heterogeneous striatal cell types based on anatomical and molecular profiles. By comparing unaffected individuals to individuals with OUD, we found significant alterations in proinflammatory, metabolic and oxidative stress, and DNA damage pathways, particularly within neurons and microglia. Additionally, we identified increased enrichment of molecular markers of DNA damage signaling that was specific to neuronal subtypes (i.e., striatal interneurons). To investigate whether opioids may directly induce accumulation of DNA damage in the brain, we assessed DNA damage markers across striatal cell types of rhesus macaques following chronic opioid administration (~6 months of twice daily opioid administration). Notably, DNA damage markers were significantly elevated across striatal neurons in opioid-exposed rhesus macaques, further implicating neuronal generated DNA damage response in response to chronic opioid use[5,7,29–33]. We also explored the interaction between OUD and sex, finding both sex-specific and cell type-specific molecular alterations in the dorsal striatum of individuals with OUD. As a resource, the filtered and annotated snRNA-seq datasets for human caudate and putamen in unaffected individuals and individuals with OUD are deposited on CZ CELLxGENE Discover Portal.

## Results

### High quality single nuclei transcriptomics identifies canonical and low abundant cell types in human dorsal striatum

To investigate the molecular features of human dorsal striatum, we collected both caudate and putamen from unaffected individuals ($n = 6$, 3 females and 3 males, for each region). Unaffected individuals were then compared to individuals with OUD to identify cell type-specific molecular alterations associated with opioid addiction. Both striatal regions were dissected from frozen postmortem tissue samples and processed to generate nuclei suspensions for snRNA-seq (Fig. 1a, b). Quality control analyses yielded a total of 98,848 nuclei for subsequent analyses—~70% of overall nuclei captured were used for analysis with an average of 5757 and 3440 nuclei per caudate and putamen, respectively, yielding an average of 8237 nuclei per individual (Fig. 1c). Across individuals, the number of nuclei captured, and the percent of nuclei analyzed were consistent ($p = 0.40$, linear regression, Supplementary Data 1-S1). Each nucleus was deeply sequenced at a 70% saturation rate, detecting an average of $3513 \pm 77$ genes (mean ± standard error) and $13,635 \pm 560$ unique transcripts per nucleus per individual (Fig. 1c). The average number of genes detected and the number of unique mapped identifiers for transcripts were consistent between unaffected individuals and individuals with OUD ($p = 0.55$, linear regression, Supplementary Data 1-S1).

We used multiple rounds of graph-based clustering, and cell label transfers from a high-quality, high-resolution snRNA-seq non-human primate striatum dataset to cluster and annotate striatal cell types[34,35] (Fig. 1d, Supplementary Data 1-S1). The first round of clustering identified major cell classes of glia and neurons (primary marker): astrocytes ($AQP4+$), endothelial cells ($CLDN+$), microglia ($CX3CR1+$), mural cells ($CD44+$), oligodendrocytes ($ASPA+$), oligodendrocyte precursors (OPC; $PDGFRA+$), and neurons ($RBFOX3+$, $GAD2+$) (Fig. 1d, e). The second round identified major striatal neuronal subtypes, including medium spiny neurons (MSN; $PPP1R1B+$) and interneurons ($LHX6+$) (Fig. 1d, e). Among MSNs, we identified various canonical subtypes, consistent with our non-human primate dataset[36], including $DRD1+$ and $DRD2+$ neurons accompanied by markers that distinguish neurochemical compartments, the striosome ($KCNIP1+$) and matrix ($EPHA4+$). We also identified less abundant neuronal subtypes including MSNs expressing both $DRD1$ and $DRD2$ (D1/D2-hybrid, known as eccentric MSNs, D1-$Pcdh8^+$, or D1H in mice)[17,34–37], mural cells, and several types of interneurons (e.g., $CCK+$, $PVALB+$, $SST+$, and $TH+$; (Fig. 1d, e)[38,39]. D1/D2-hybrid MSNs expressed conserved marker genes largely distinct from D1- and D2-MSNs (Fig. 1e, Supplementary Data 1-S2). These D1/D2-hybrid MSNs co-express both $DRD1$ and $DRD2$ transcripts at the single cell level (Fig. S3), consistent with in situ mRNA hybridization and single cell RNA-seq data from both mice[35,37,40] and non-human primates[34]. The average per cell type metrics showed more genes and unique transcripts in neurons (Number of genes$_{Neurons}$ = 7318 ± 125; UMsI$_{Neurons}$ = 44,585 ± 1,301), compared to glial cells (Number of genes$_{Neurons}$ = 2911 ± 74; UMsI$_{Neurons}$ = 8433 ± 349; Fig. S4). Further, proportions of low or high abundance striatal cell types were consistent across brain regions, between sexes, and between unaffected individuals and individuals with OUD (FDR = 0.25, mixed effects linear regression; Fig. 1f; Supplementary Data 1-S2). Overall, we captured a high-quality snRNA-seq dataset of postmortem human caudate and putamen that is deeply sequenced and annotated for in-depth investigations into OUD.

### Neuronal and glial cell type-specific expression of opioid receptors in human striatum

Opioids, including endogenous opioids, enkephalin, dynorphin, and β-endorphin, differentially activate several classes of opioid receptors, including mu, delta, and kappa, encoded by $OPRM1$, $OPRD1$, and $OPRK1$, respectively. Previous efforts have mapped striatal expression of opioid receptors in non-human primates and rodents, characterizing species-specific and cell type-specific patterns of expression[41,42]. We investigated the expression of opioid receptors and cognate ligands within the dorsal striatum. First, $OPRM1$ was detected in each MSN subtype, with the highest level of expression in D1/D2-hybrid and D2-striosome MSNs (Fig. S5). We reproducibly detected $OPRM1$ in microglia across individuals and biological samples, consistent with findings in rodents[32,43], establishing a potential pathway for opioids to directly modulate neuroimmune signaling[5]. D1-striosome MSNs expressed the gene encoding the preproprotein, prodynorphin ($PDYN$), while D2-matrix and D2-striosome MSNs co-expressed $OPRD1$ and proenkephalin ($PENK$) at markedly higher levels than other striatal cell types (Fig. S5). Preferential expression of $OPRD1$ and $PENK$ in D2 MSNs in human striatum is consistent with rodent striatum[44]. Across MSNs, we detected $OPRK1$ expression at many folds lower than $OPRM1$, $OPRD1$, $PENK$, or $PDYN$, suggesting that targets of the dynorphin-kappa opioid receptor signaling by D1-striosome MSNs may primarily be outside of dorsal striatum[45,46]. Between unaffected individuals and individuals with OUD, we found no significant differences in opioid receptor and/or endogenous ligand expression within specific cell types. Other mechanisms involved in the post-transcriptional and/or post-translational regulation of opioid receptor and ligand expression, along with receptor binding activity, may be altered in OUD.

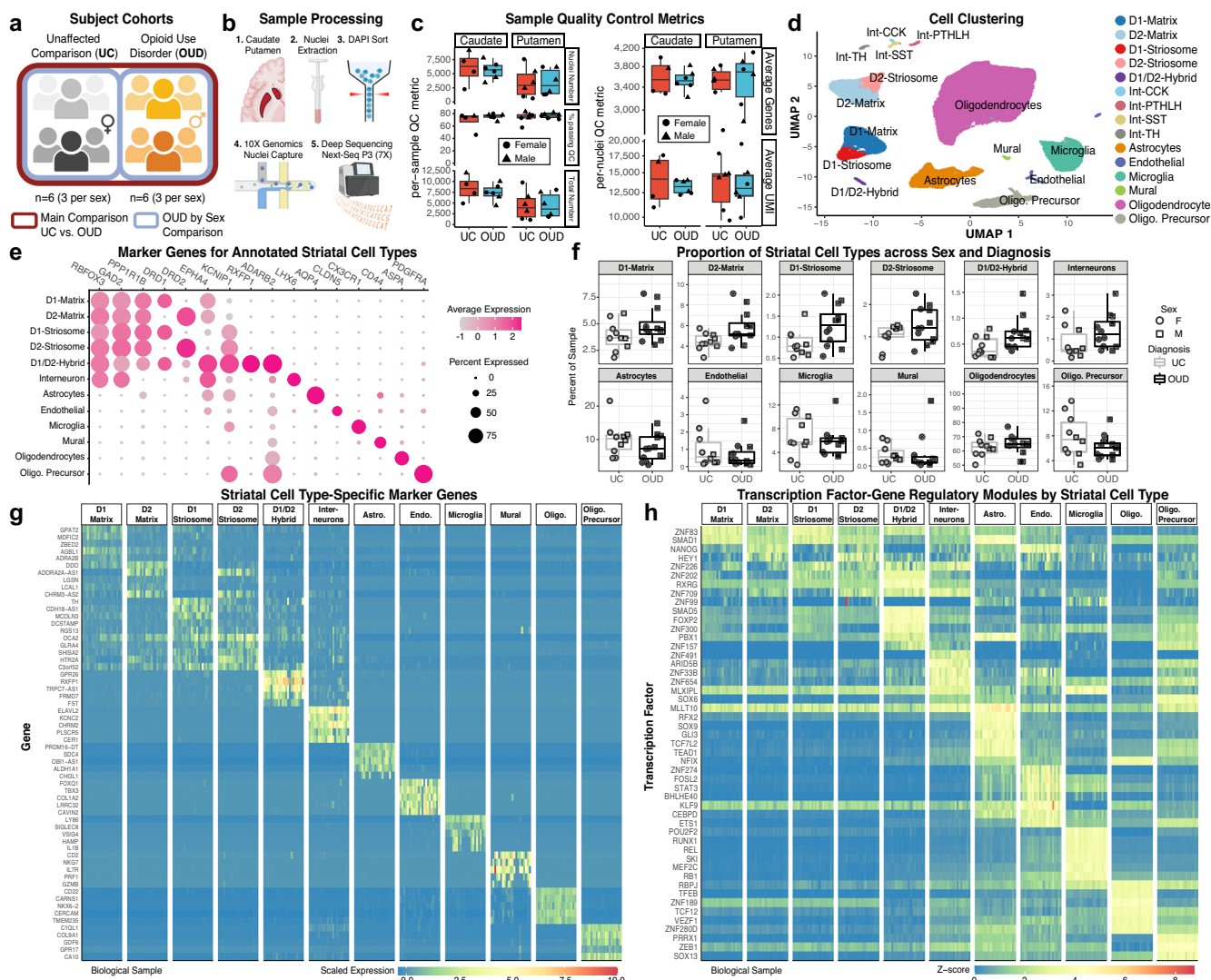

**Fig. 1 | Single nuclei RNA-sequencing of postmortem brain to identify specific cell types in human striatum. a** Post-mortem human brain cohort design and analysis to compare unaffected comparison (UC) individuals with individuals with opioid use disorder (OUD). **b** Schematic of the sample collection process to isolate single nuclei and generate single nuclei RNA-sequencing libraries in balanced batches from the caudate and putamen of human postmortem striatum. **c** Per-sample quality control (QC) metrics and average per-nuclei QC metrics across tissue sources and diagnoses. Each data point consists of $N = 22$ caudate and putamen samples from $M = 12$ individuals. **d** Single nuclei clustering and label annotation of dorsal striatal cell types using a high-quality non-human primate reference dataset[36]. A low dimensionality projection of striatal cell types after QC filtering and annotation. **e** Dot plots of the marker genes used to annotate the various cell types of the dorsal striatum based on a non-human primate reference. The normalized expression patterns are averaged across all cells and individuals. **f** Boxplot showing the relative percent of each cell type detected in each biospecimen that passes QC. No significant differences were observed between UC individuals and individuals with OUD. Each data point consists of $N = 22$ caudate and putamen samples from $M = 12$ individuals. **g** Cell type-specific marker genes and (**h**) marker transcription factor-gene regulatory networks identified in cell types of human striatum. Schematics in (**a**) and (**b**) created using BioRender.com. Source data are provided as a Source Data file. Boxplots in 1 C and 1 F are plotted as median, the 25% and 75% percentiles, and non-outlier maxima and minima.

## Identification of cell type-specific transcription factor and gene regulatory networks in human dorsal striatum

Following cell annotation, we aimed to identify putative regulatory transcription factor-gene regulatory networks across striatal cell types. First, we identified marker genes for each of the annotated cell types that were reproducible across biological replicates (mean ± standard error, 722 ± 172 specific marker genes per cell type; Figs. 1d, e, g, S1; Supplementary Data 1-S3). Glial cell types yielded more marker genes (1134 ± 204) compared to neuronal cell types (310 ± 131). Next, we inferred gene regulatory networks directly from each of the cell type-specific marker datasets using machine-learning, SCENIC[47]. SCENIC is based on the premise that transcriptional regulation of gene expression is, in part, based on transcription factor binding to proximal gene promoters. Based on single cell gene expression and the enrichment of promoter binding sites, we used SCENIC to build transcription factor

gene regulatory modules that were specific to each of the neural cell types identified in human striatum. Collectively, we identified many transcription factors and gene regulatory modules per cell type (47 ± 15 specific modules; glial subtypes, 90 ± 19; neuronal subtypes, 12 ± 6; Fig. 1h; Supplementary Data 1-S4). For example, ZNF83 (HPF1) transcription factor-regulatory module was enriched in D1-matrix and D1-striosome MSNs, known to be involved in DNA damage repair. RXRG[28,48,49] and FOXP2[50] transcription factor-regulatory modules were highly enriched in D1/D2-hybrid MSNs relative to other MSN subtypes, both of which are implicated in psychiatric disorders, including substance use[51]. ZNF202 was also highly enriched in D1/D2-hybrid MSNs−transcription factor involved in cellular metabolism and the direct modulation of dopamine receptor 3 (DRD3). Other modules included SOX9[52] and TCF7L2[53] enriched in astrocytes, and others in microglia, such as RUNX1, REL, and MEF2C, with the RB1 and NFIX[54]

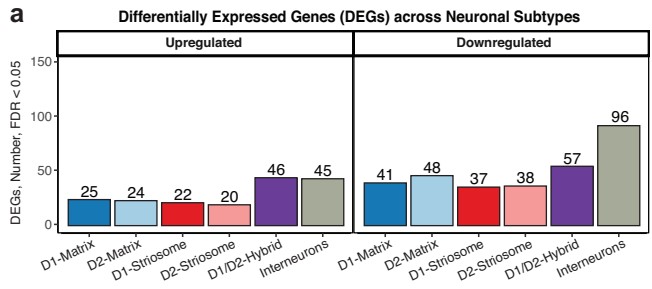

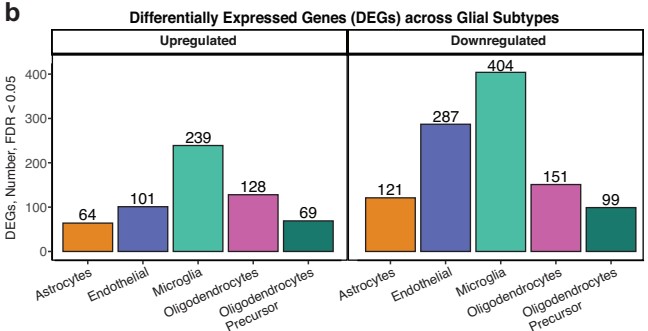

**Fig. 2 | Differentially expressed genes in human striatal cell types associated with OUD. a** Barplot of differentially expressed genes (DEGs) either upregulated or downregulated in striatal neurons at a false discovery rate less than 0.05 (FDR < 0.05). **b** Barplot of DEGs either upregulated or downregulated in striatal glia at FDR < 0.05. Source data are provided as a Source Data file.

modules enriched in oligodendrocytes (Fig. 1h). Together, the annotated cell type clusters in human dorsal striatum, their associated marker genes, and related transcription factor gene regulatory modules offer insights into the molecular signaling pathways that contribute to the heterogeneity of cellular identity in human striatum.

## Differentially expressed genes in specific striatal cell types associated with OUD

Similar to our prior work[2,4,5], we assembled a cohort of individuals diagnosed with OUD with a long history of opioid use (time since clinical diagnosis of at least four years), who also deceased from opioid overdose. Individuals with OUD were matched with unaffected individuals based on sex, age, postmortem interval (PMI), and RNA integrity number (RIN) (Supplementary Data 1-S5). Gene expression was aggregated across individual cell types to generate pseudobulk profiles for each major cell type. To identify differentially expressed genes (DEGs), gene expression profiles for each major cell type were compared between unaffected individuals and individuals with OUD, while covarying age, PMI, RIN, cell type abundance, gene detection rate, and surrogate variables. We identified 1,765 DEGs across cell types (197 ± 43 genes per cell type; FDR < 0.05; Fig. 2; Supplementary Data 1-S6), with more DEGs in glial cells relative to neurons (Fig. 2a, b).

Using gene set enrichment analyses (GSEA)[55], we identified pathways significantly enriched in aggregate cell types and across specific cell types. Across cell types, we found 286 pathways (34 ± 7 pathways per cell type; FDR < 0.05, Supplementary Data 1-S7) and 51 differentially regulated transcription factor regulatory modules (5.1 ± 0.7 per cell type, FDR < 0.05, Supplementary Data 1-S8). The cell type-specific transcriptional differences in individuals with OUD were represented largely by distinct biological processes between neurons and glia (Fig. S6). Among MSNs, DEGs tended to be shared between MSN subtypes, while DEGs tended to be more distinct between glial subtypes and between interneuron subtypes (Fig. S5). We identified a higher number of DEGs in interneuron subtypes and D1/D2-hybrid MSNs compared to canonical D1-MSNs or D2-MSNs. DEGs in D1/D2-hybrid MSNs were unique relative to canonical MSNs, suggesting

subpopulations of striatal MSNs exhibit different molecular responses to chronic opioid use and other factors associated with OUD (Fig. S5).

## Enriched pathways among neurons converge on various pathways of cell stress in OUD

We conducted separate pathway analyses on upregulated and downregulated transcripts (Fig. 3a) between unaffected individuals and individuals with OUD. In neurons, most of the upregulated pathways in OUD were related to processes of cell stress response (Fig. 3c). For example, pathways of DNA replication and cell cycle re-entry were upregulated in MSNs (Fig. 3c; Supplementary Data 1-S9)[56]. Both *MCM8* (log2FC > 1.06, FDR < 0.041) and myosin light-chain 6, *MYL6*, (log2FC > 1.13, FDR < 0.045), genes involved in DNA replication, were significantly upregulated in striatal MSNs (Fig. 3a). MCM8, forms a complex with MCM9 to facilitate repair of double-stranded DNA breaks[57,58]. MYL6 is a key factor in rho-GTPase signaling, known to facilitate glutamate receptor endocytosis[59] and neuroplasticity[60,61], and may be involved in DNA damage via the activation of RAC1[62]. *APOE*[63–65] (striosome MSNs and D2-matrix MSNs: log2FC > 2.50, FDR < 0.025; Fig. 3a) and *GPX4* (D1-striosome MSNs: log2FC > 1.17, FDR < 0.044) were also upregulated in MSNs of individuals with OUD. Both *APOE* and *GPX4* are involved in neuronal oxidative stress, where GPX4 buffers the accumulation of reactive oxygen species to prevent apoptosis[63–65]. Accumulation of reactive oxygen species and oxidative stress in neurons involves alterations in redox signaling and mitochondrial respiration[66]. Indeed, across multiple MSN subtypes, genes involved in mitochondrial respiration were significantly upregulated in OUD, including a complex III subunit of the mitochondrial respiratory chain, *UQCR11* (log2FC > 1.36, FDR < 0.042), and a subunit of complex I, *NDUFA4* (log2FC > 1.29, FDR < 0.030).

We identified several transcription factor gene regulatory modules that were significantly upregulated in different neuronal subtypes, including cell senescence, DNA damage, and inflammation, and stress (Fig. 3b, e; FDR < 0.05). For example, the module linked to BCLAF1 was upregulated in OUD, a transcription factor involved in the transduction of NFkB-dependent signaling and the activation of DNA-damage-induced senescence[39,67,68] (Fig. 3e). Another module was associated with the upregulation of the neuron-specific glucocorticoid receptor transcription factor, NR3C1, previously linked to the impact of psychosocial stress on the human brain[69,70] (Fig. 3e). Other stress-related transcription factor gene regulatory modules were upregulated in OUD, including ATF2[71] and ZNF518A. Both transcription factor modules were upregulated primarily in D1-matrix and D2-matrix MSNs (FDR < 0.04). ATF2 is induced by stress in mouse dorsal striatum[71] and phosphorylated downstream of delta opioid receptor activation[72], suggesting delta opioid receptor activation of ATF2-dependent transcription in striatal matrix MSNs[73] may be due to an interplay between stress and opioids in OUD.

Compared to upregulated modules, we identified almost twice as many significantly downregulated transcription factor gene regulatory modules in individuals with OUD (Fig. 3d). We identified several downregulated pathways involved in neuroprotection (Fig. 3d). For example, the humanin-like 8 gene, *MTRNR2L8*, was significantly downregulated in OUD across every neuronal subtype (log2FC < −3.79, FDR < 0.0054). *MTRHR2L8* and the mitochondrial paralog, *MTRNR2*, act as neuroprotective factors in response to neurodegeneration and the induction of DNA damage[74,75]. The top DEGs within clusters of pathways involved in UV irradiation, another potential link to DNA damage (FDR < 0.029, Fig. 3d), included *FOXO1*, *TLE4*, *SHOC2*, *MEIS2*, and *THBS1* (log2FC < −0.64, FDR < 0.0422) in interneurons and CLASP1, PIK3C3, and MTUS1 (log2FC < −0.64, FDR < 0.042) in D1/D2-hybrid MSNs. Additionally, downregulation of MEF2B and BACH1 transcription factor gene regulatory modules were found in D1-matrix MSNs in OUD (Fig. 3e), accompanied by downregulation of MAF, EST1, ETV1, and PRRX1

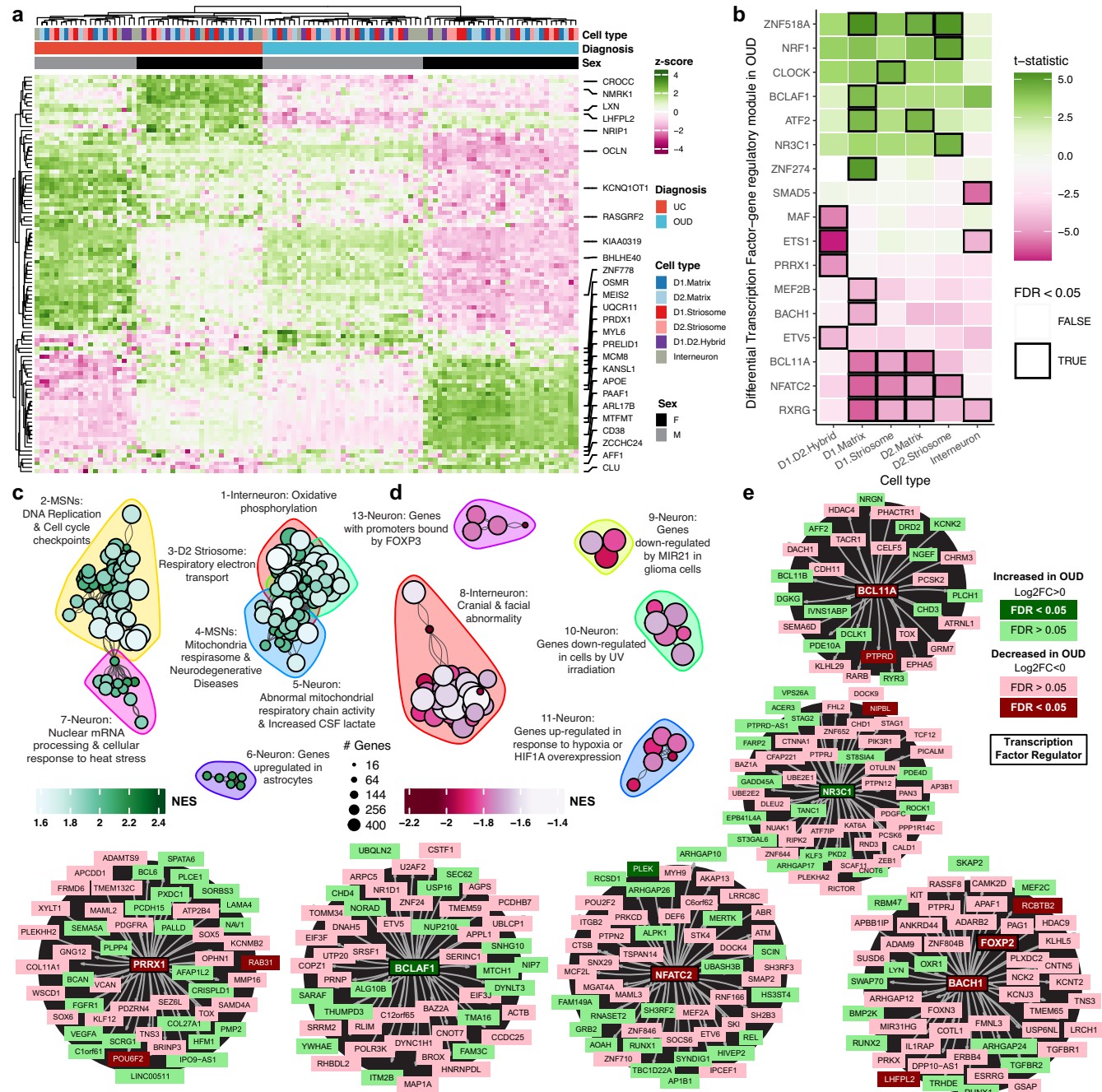

**Fig. 3 | Transcriptional alterations in specific neuronal cell types in human striatum associated with OUD. a** Expression heatmap of z-normalized pseudobulk gene expression, where the rows are differentially expressed genes (DEGs) and the columns are the biological replicates labeled by cell type, diagnosis, and sex of each individual. Genes that are represented by the enriched pathways in (**c, d**) are labeled along the right margins. **b** A heatmap plotting the t-statistic of differentially activated transcription factor gene regulatory networks across neuronal subtypes in opioid use disorder (OUD). **c, d** Network plot of significant clusters and enriched pathways, where each point is a significantly enriched pathway in a neuronal cell type and lines represent the proportion of shared genes between two pathways.

Each point is colored by the normalized enrichment score (NES), indicating whether a pathway is enriched in upregulated or downregulated genes. The color outlines represent the unique clusters of interconnected pathways by shared genes and the nearby text labels the unique cluster number and briefly summarizes which cell types and pathways are represented. **e** Cluster maps of select transcription factor gene regulatory networks. Each transcription factor or gene is colored to denote the direction and significance of differential expression in neuronal subtypes in OUD. Source data are provided as a Source Data file. Source data are provided as a Source Data file.

---

modules in D1/D2 hybrid MSNs (FDR < 0.043), each of which are implicated in neuronal stress[76,77]. Notably, BACH1, BCL11A, and PRRX1 have downstream targets with concordant differential expression in OUD (*e.g., FOXP2, POUF3, PTPRD, RAB31, RCBTB2*; Fig. 3e), indicating coordinated changes in striatal gene networks implicated in neuronal stress associated with opioid addiction.

Additional findings included modules for NFATC2 and RXRG (FDR < 0.042; Fig. 3e). Both nuclear factor of activated T-cell C2 (NFATC2) and the retinoid X receptor gamma (RXRG) of the RXR family of receptors link changes in neuronal activity associated with addictive drugs to processes involved in neuroprotection, neurodegeneration[78,79] and reward-related behaviors[49,80].

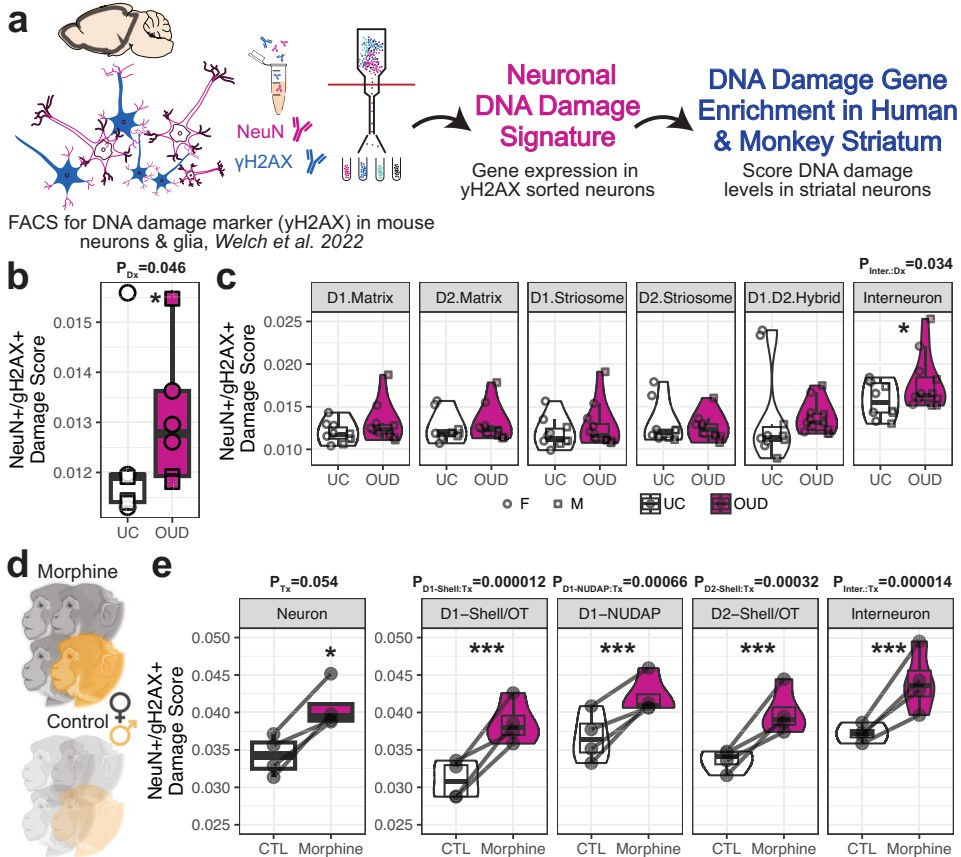

**Fig. 4 | Elevated markers of DNA damage in striatal neurons associated with OUD and chronic morphine in rhesus macaque. a** Schematic of the application of DNA damage gene signatures from a mouse model of Alzheimer's Disease[13] to score DNA damage in human and rhesus macaque striatal neurons. **b** Boxplot of individual-level pseudobulk average DNA damage scores across striatal neurons between unaffected individuals and individuals with opioid use disorder (OUD) ($P_{Dx}$ = 0.046, two-sided linear regression, 15 degrees of freedom). Each data point comes from $N$ = 22 biologically independent samples of caudate and putamen from M = 12 individuals. **c** Violin-boxplot of cell type-level pseudobulk average DNA damage scores between unaffected individuals and individuals with OUD. The significant cell type interaction effect with diagnosis two-sided P-values from one linear regression (two-sided linear regression, 114 degrees of freedom). Each data point comes from each neuronal cell type from $N$ = 12 biologically independent individuals. **d** Schematic of chronic morphine exposure or unexposed rhesus macaques (N = 4 individuals per treatment). **e** Boxplots of individual level pseudobulk average DNA damage scores across striatal neurons or neuronal subtypes between control individuals or those exposed to chronic morphine. The effect of morphine in all striatal neurons is one linear regression (two-sided linear regression, 2 degrees of freedom). The morphine by cell type interaction effect for each striatal neuron subtype is another linear regression (two-sided linear regression, 20 degrees of freedom). The significant effect of chronic morphine treatment P-values from linear regressions are reported above each plot. (*$P < 0.05$; **$P < 0.01$; ***$P < 0.001$). Each data point comes from all neuronal cell types or each neuronal cell type from $N$ = 8 biologically independent *rhesus macaque* striatal samples. Schematics in (**a**, **d**) created using BioRender.com. Source data are provided as a Source Data file. Boxplots in 4b-c and 4e are plotted as median, the 25% and 75% percentiles, and non-outlier maxima and minima.

## Enrichment of DNA damage markers in specific neuronal subtypes in OUD

Several pathways and key transcription factor modules were related to processes of DNA damage and repair processes in individuals with OUD. Based on our results and other findings[11], we further investigated the relationship between DNA damage and OUD in specific cell types (Fig. 4a). In striatal neurons, we observed a significant enrichment of DNA damage markers in individuals with OUD ($p$ = 0.046; linear regression t = 2.18; Fig. 4b). More specifically, a significant increase in DNA damage markers were found in interneurons of individuals with OUD ($p$ = 0.035, linear regression t = 2.14; Figs. 4c, S6; Supplementary Data 1-S10). We further resolved the augmentation of DNA damage markers among striatal interneuron subtypes in OUD, except for *PTHLH*+ interneurons (Fig. S7, p < 0.025). Our findings suggest striatal neurons may incur DNA damage in response to opioids and other factors, such as stress, neuroinflammation, and hypoxia (Fig. 3), associated with OUD and opioid overdose. Interestingly, hypoxia pathways were significantly downregulated in OUD (Fig. 3d; Supplementary Data 1-S9), including decreased expression of genes that are activated

in response to hypoxic events (*e.g., NMRK1, BHLHE4O*; Fig. 3a)[81], and overexpression of hypoxia-inducible transcription factor, HIF1A. Therefore, while respiratory depression is a hallmark of opioid overdose, downregulation of hypoxia-responsive genes points toward possible compensation in neurons secondary to periodic hypoxia and altered redox states, consistent with rodent models of substance use[82].

Individuals with OUD were selected based on time since initial diagnosis of at least four years and other key factors, capturing the consequences of long-term opioid use on the human brain. Individuals with OUD also died of accidental opioid overdose, which leads to the possibility of introducing the impact of acute opioids and other substances on the brain. To further investigate the consequences of long-term opioid use on DNA damage-related processes in the brain, we assessed DNA damage marker enrichment in striatal neurons of non-human primates following chronic opioid administration using snRNAseq (Fig. 4d). Male and female rhesus macaques were administered morphine for ~6 months, twice daily, then the striatum was rapidly dissected, stored, and later processed for nuclei extraction (Figs. 4d, S8A, B, Supplementary Data 1–S20). Individual rhesus

macaques were matched on age, sex, and body weight between vehicle and morphine treated groups (Fig. S8A). Cell types were clustered across individual rhesus macaques, which yielded similar results to our previous non-human primate striatal findings[34] (Fig. S8C). Consistent with our findings of elevated DNA damage markers in individuals with OUD, rhesus macaques administered morphine for nearly 6 months exhibited significant enrichment of DNA damage markers in striatal neurons ($P$ = 0.05, linear regression; Fig. 3e). DNA damage marker enrichment was found across each of the major neuronal subtypes ($P$ < 0.00066, linear regression; Fig. 3e). Despite limitations (i.e., species differences of striatal regions and species-specific opioid-induced gene expression changes) (Figs. S8, S9)[83], elevated DNA damage markers were evident in striatal neurons from individuals with OUD and non-human primates chronically administered opioids.

### Enriched pathways among glia support molecular signatures related to neuroinflammation and synaptic signaling in OUD

Relative to neurons, we identified more than twice as many DEGs in glial cells between unaffected individuals and individuals with OUD (Figs. 2b, 4a). The number of DEGs were independent of cell type proportions and sequencing depth, as microglia and endothelial cells had the most DEGs among glial cells, and expression patterns were largely distinct between glial cell types (Fig. S6). An exception was the robust enrichment of genes involved in interferon response in multiple glial cell types, including astrocytes and oligodendrocytes (Fig. 5c). Interferon response genes were significantly upregulated in all glial cell types in individuals with OUD (FDR < 0.047, Supplementary Data 1–S11), with the following interferon-related hub genes: *CMP2*, *HLA-F*, *IFI44*, *PPM1B*, and *RSAD2* (Fig. 5a). We also observed an upregulation of the NFKB1 transcription factor gene regulatory module in astrocytes (Fig. 5b, e), and the upregulation of the STAT3 module in oligodendrocyte precursor cells (OPC,FDR < 0.036; Fig. 5b). Several of the predicted NFKB1 gene targets were also differentially expressed including the inhibitor of this pathway, *NFKBIA* (log2FC = 1.86, FDR = 0.039; Fig. 5e). Other glial cell types showed non-significant activation of this module, with the weakest being microglia. In response to cell stress, microglia release pro-inflammatory cytokines that may trigger an interferon response in other cell types, potentially responsible for broader patterns of interferon activation we observe across multiple striatal cell types in OUD[84].

Altered neuroinflammatory signaling in microglia has previously been reported in postmortem brains from individuals with OUD[5,9]. Several DEGs in microglia lend further support roles for inflammation in synaptic plasticity in OUD. For example, the PAX-FOXO1, CDH1, and neuron projection signaling pathways were enriched among microglia DEGs (FDR < 0.049). PAX-FOXO1 signaling regulates cellular senescence in the brain and is involved in the pathogenesis of several neurodegenerative disorders[85], while CDH1 signaling is involved in glial cell migration and axonal projections[86]. In the neuronal projection pathway, *ADGRB3* was among the top downregulated genes in microglia of individuals with OUD (FDR = 3.5e-7). *ADGRB3* (known as BAI3, brain-specific angiogenesis inhibitor 3) encodes for a protein that has high-affinity for complement, C1q, an innate immune component released by neurons to eliminate damaged or inactive synapses[87]. Downregulation of *ADGRB3* in microglia may lead to aberrant synaptic formation.

In support of synaptic changes associated with opioid addiction, several genes involved in glutamatergic and GABAergic neurotransmission were altered in individuals with OUD (astrocytes: metabotropic glutamate receptor 5, *GRM5*, log2FC = −3.13, FDR = 1.06 E −14 and glutamate ionotropic receptor AMPA type subunit 1, *GRIA1*, log2FC = −1.98, FDR = 2.8 E −8; oligodendrocytes: GABA type A receptor subunit beta1, *GABRB1*, log2FC = −0.717, FDR = 0.023; microglia: GABA type A receptor subunit gamma 2, *GABRG2*, log2FC = −1.45, FDR = 0.038). Indeed, we identified several pathways related to various

synaptic (e.g., glutamatergic synapse) and immune functions (e.g., TNF-alpha signaling) in gene co-expression network modules unique to MSN subpopulations and microglia (Figs. S10–12; Supplementary Data 1–S18, S19). Together, our cell type-specific findings suggest an interplay between microglial-dependent signaling and synaptic plasticity in OUD, consistent with our previous bulk transcriptomics findings in striatum from individuals with OUD[5]. With single nuclei resolution, we find the elevation of neuroinflammatory pathways related to microglial activation in OUD is likely due to transcriptional alterations within microglia, rather than pronounced changes in the number of striatal microglia in individuals with OUD (FDR = 0.48, linear mixed effect regression for differential abundance; Fig. 1f; Supplementary Data 1–S2).

Endothelial cells displayed the second most DEGs in OUD. Upregulated pathways in endothelial cells included several growth factors, VEGFR, EGFR, and PDGFR[88,89] (Fig. 5c). Several of these growth factors are linked to nociception and opioid tolerance[89,90]. Other pathways included leukocyte chemotaxis and dendrite morphogenesis. Among the densely connected endothelial pathways, the top significantly upregulated hub gene was the brain-specific angiogenesis inhibitor 1-associated protein 2, *BAIAP2* (log2FC = 2.31, FDR = 0.0017), and, interestingly, the Alzheimer's related microtubule-associated protein tau, *MAPT* (log2FC = 1.58, FDR = 0.036). The upregulation of *MAPT* may contribute to neurovascular brain insults in OUD[6,91].

The identification of differentially expressed transcription factor gene regulatory modules across glial cell types highlights several key factors in the regulation of immune response related to OUD. In oligodendrocyte precursor cells, the CEBPB module was upregulated in OUD (FDR = 0.031; Fig. 5b, e), with prior studies showing diverse functions of CEBPB in proinflammatory states[92] and the direct regulation of *APOE* in Alzheimer's disease-related pathologies[93,94]. Notably, *APOE* is concordantly upregulated in oligodendrocyte precursor cells (log2FC = 2.56, FDR = 5.9e-4), along with other predicted CEBPB targets (Fig. 5b, e). The RXRG transcription factor-gene regulatory module, which was downregulated in neuronal subtypes, was also downregulated in microglia (Fig. 5b, e, FDR = 0.027). An inferred target of RXRG is *FOXP2* (Fig. 5e), which was recently associated with OUD and other substance use disorders at the genome-wide level[95]. FOXP2 expression in microglia may be unique to humans compared to other primates. Thus, downregulation of the FOXP2 transcription factor gene regulatory module in human microglia may link the genetic risk of OUD to other addiction risk traits[96,97].

### Sex-specific transcriptional alterations across striatal cell types associated with OUD

Prevalence rates of OUD and responses to opioids are dependent on sex[98–100]. Between unaffected individuals and individuals with OUD, we found a significant effect of sex on DEGs in neurons and glia. Overall, the impact of sex on DEGs in OUD was magnified in neurons (Fig. 3a) relative to glia (Fig. 5a). Therefore, we conducted complementary, secondary analyses to identify sex-specific transcriptional alterations in neurons and glia associated with OUD (Supplementary Data 1–S12). First, we identified DEGs within either female or male individuals with OUD (Fig. 6a, FDR < 0.05), revealing more sex-specific DEGs in glial cells relative to neurons. Additionally, we found a higher number of DEGs in females than males, (360 ± 60 versus 212 ± 67, respectively), in glial cells, and similarly in neurons (females: 91 ± 9.1; and males: 66 ± 7.9. Second, we explored the interaction between sex and OUD across genes and cell types to identify cell type-specific gene expression changes occurring in only males or females of individuals with OUD. Consistent with our analysis on the main effect of sex in OUD, we found more DEGs in glia (291 ± 55) than neurons (97 ± 5.6) with sex-specific changes in OUD (FDR < 0.05, Supplementary Data 1–S12). Among glial cell types, more DEGs were identified in females with OUD relative to males (Fig. S13). The complete set of gene alterations

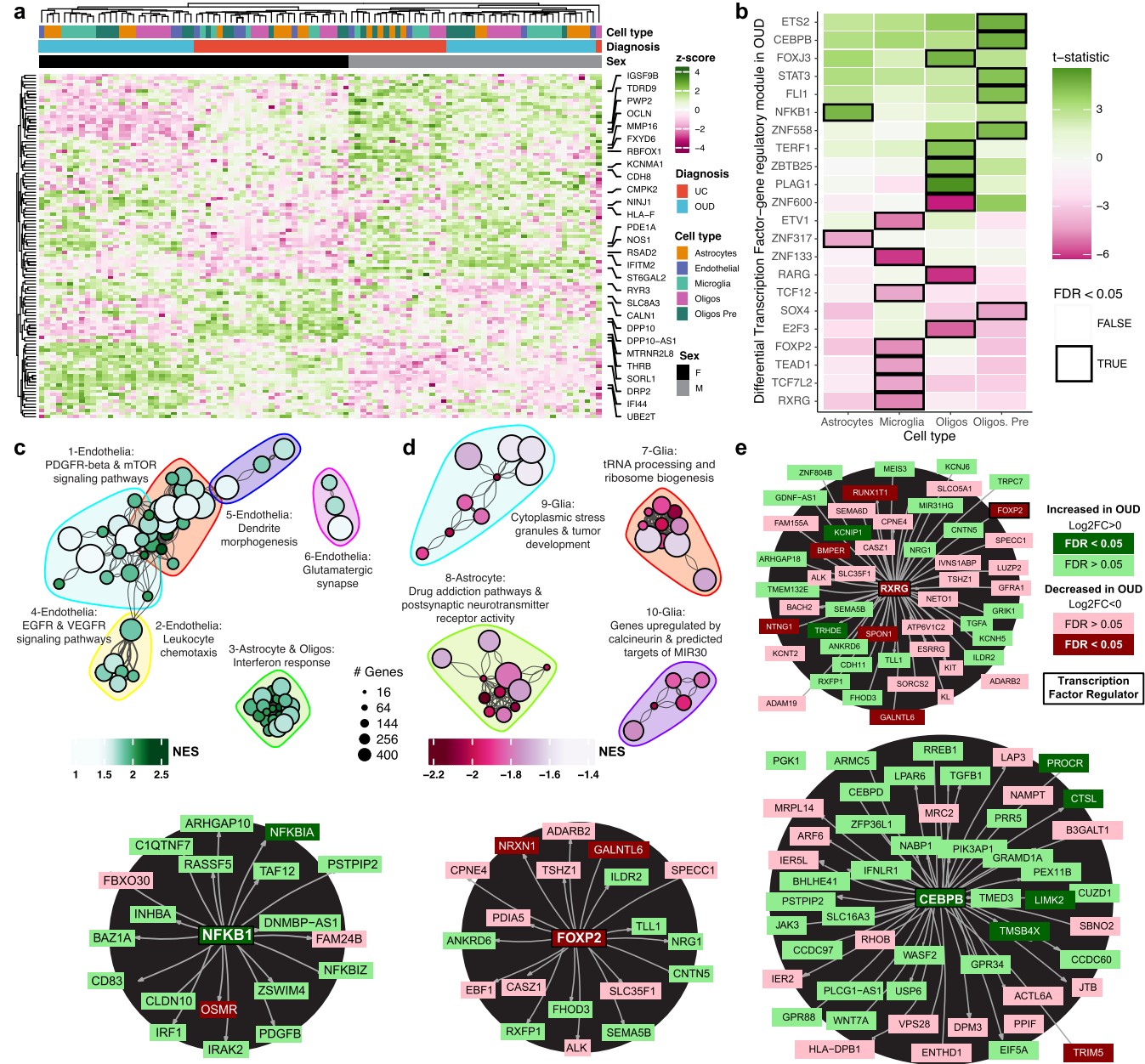

**Fig. 5 | Transcriptional alterations in specific glial cell types in human striatum associated with OUD. a** Expression heatmap of z-normalized pseudobulk gene expression, where the rows are differentially expressed genes (DEGs) and the columns are the biological replicates labeled by cell type, diagnosis, and sex of each individual. Genes that are represented by the enriched pathways in (**c**, **d**) are labeled along the right margins. **b** A heatmap plotting the t-statistic of differentially activated transcription factor gene regulatory networks across glial subtypes in opioid use disorder (OUD). **c**, **d** Network plot of significant clusters and enriched pathways, where each point is a significantly enriched pathway in a glial cell type and lines represent the proportion of shared genes between two pathways. Each point is colored by the normalized enrichment score (NES), indicating whether a pathway is enriched in up- or down-regulated genes. The color outlines represent the unique clusters of interconnected pathways by shared genes and the nearby text labels the unique cluster number and briefly summarizes which cell types and pathways are represented. **e** Cluster maps of select transcription factor gene regulatory networks. Each transcription factor or gene is colored to denote the direction and significance of differential expression in glial subtypes in OUD. Source data are provided as a Source Data file.

associated with OUD and sex are reported in Supplementary Data 1–S12. Lastly, we identified DEGs that were different between females and males within unaffected individuals and individuals with OUD. As expected, sex-specific DEGs in individuals with OUD compared to unaffected individuals were largely different, suggesting gene alterations dependent on sex and diagnosis were independent of naturally occurring gene expression variations between sexes (Fig. S14). Collectively, our findings indicated that many genes altered in OUD depend on sex across striatal cell types, especially across subtypes of glial cells.

To investigate which biological processes underlie sex-specific changes in OUD, we performed pathway analyses using female-biased and male-biased DEGs. In female individuals with OUD, pathways were primarily associated with the upregulation of DNA repair processes, particularly in MSNs, accompanied by upregulation of interferon response signaling in glial cells, along with the upregulation of synaptic-related functions in astrocytes (Fig. 6b; Supplementary Data 1–S13, S15). We then investigated significant key sex-specific factors in specific cell types of individuals with OUD. We identified the FK506 binding protein 5, *FKBP5*, as a potential factor in the role for sex in opioid use.

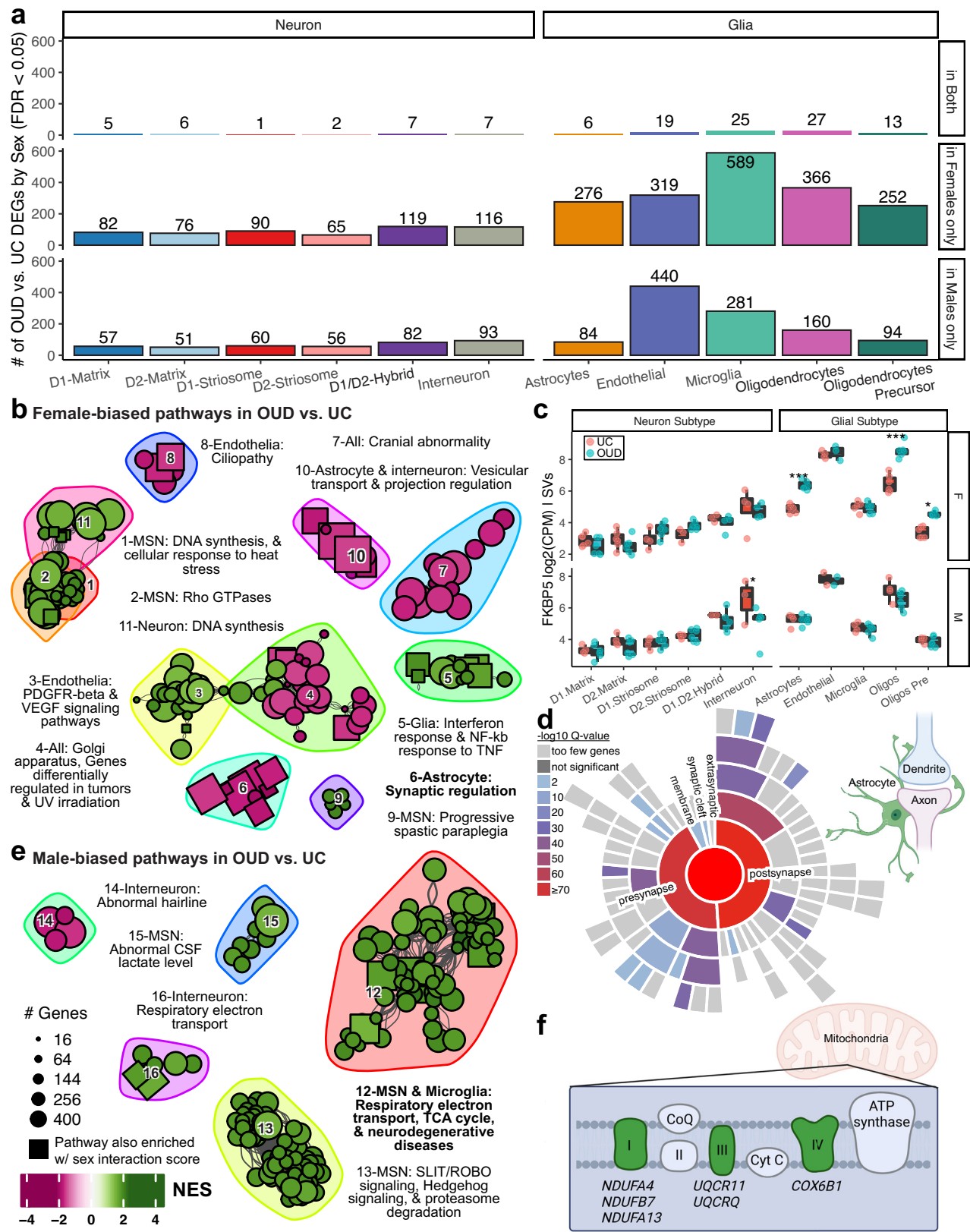

In the brain, FKBP5[101,102] is likely a key factor involved in stress and opioids in females. *FKBP5* was significantly upregulated in astrocytes, oligodendrocytes, and oligodendrocyte precursor cells of females with OUD (Fig. 6c; log2FC$_{F \, glia}$ > 1.30, FDR$_{F \, glia}$ < 0.0165, log2FC$_M$ = −1.62, FDR$_{M \, interneurons}$ < 0.040). Preclinical studies have found marked sex differences of *Fkbp5* expression in dorsal striatum in response to opioids[103,104], and thus, investigated whether there were similarities in gene changes between rodents and humans in dorsal striatum. Indeed, gene sets from rodents exposed to opioids were significantly enriched for genes we found in both glia and oligodendrocytes of females with OUD (FDR$_{F \, glia}$ < 0.027; Supplementary Data 1-S16). Cross-species enriched DEGs were associated with reduced

**Fig. 6 | Sex-biased transcriptional alterations in striatal cell types associated with OUD. a** Barplot showing the number of differentially expressed genes (DEGs) detected by cell type in sex-specific differential analyses comparing unaffected individuals and individuals with OUD. The bar plots show significant DEGs in only female individuals, in only male individuals, or in both groups (FDR < 0.05).
**b** Network plot of significant clusters and enriched pathways as in Figs. 2b, c and 4b, c. Here, enriched pathways include only DEGs calculated within female individuals if less significant than those calculated within male individuals. Pathways with a square point are also enriched using an alternate calculation of a sex-interaction score. **c** Boxplots showing sex- and cell type-specificity of differential expression of *FKBP5*. The boxplot labels the log2(counts per million) normalized gene expression values regressing out covariates and surrogate variables. Each point is a biological replicate colored by diagnosis (*$P_{Dx\,within\,Sex}$ < 0.05; **$P_{Dx\,within\,Sex}$ < 0.01; ***$P_{Dx\,within}$

$_{Sex}$ < 0.001, two-sided limma regression, exact P-values reported in Supplementary Data 1–S12). Each data point consists of $N = 22$ caudate and putamen samples from $M = 12$ individuals. **d** A sunburst plot representing curated synaptic gene sets from SynGO from genes enriched in cluster 6, a female-biased set of pathways enriched in astrocytes. **e** A network plot as in (**b**); however, displaying the male-biased pathways. **f** Diagram showing the male biased DEGs that are components of the electron transport chain, including genes involved in mitochondrial functions (*NDUFA4*, *NDUFB7*, and *NDUFA13* of complex I of mitochondrial respiratory chain; *UQCR11* and *UQCRQ* of complex II; and *COX6B1* of complex IV). Schematics in (**f**) created using BioRender.com. Source data are provided as a Source Data file. Boxplots in 6 C are plotted as median, the 25% and 75% percentiles, and non-outlier maxima and minima.

expression of genes involved in synaptic regulation, particularly in astrocytes (Fig. 6b; cluster 6), spanning both pre- and postsynaptic compartments (Fig. 6d; Supplementary Data 1–S14, S15). In contrast, male-biased molecular alterations in OUD were largely found in microglia, MSNs, and interneurons (Fig. 6e; clusters 12, 15, 16), enriched in mitochondrial pathways (Fig. 6f). Collectively, our evidence of sex differences in cell type-specific molecular signaling suggests an augmented, magnified glial cell response in females compared to males with OUD.

## Discussion

Using single nuclei transcriptomics technologies, we identified both canonical neuronal and glial cell types in human striatum (e.g., D1-MSNs and D2-MSNs), along with several of the less abundant cell types, including subclasses of striatal interneurons and D1/D2-hybrid neurons. We also characterized the expression of specific opioid receptor and endogenous ligand subclasses across cell types, highlighting expression patterns unique to human striatum. Our single nuclei findings provide further insights into the specific cell types that may be impacted in OUD, while highlighting several putative mechanisms in human striatal neuronal and glial subtypes. A common theme that emerged from our analyses was the involvement of processes related to neuroinflammation and cell stress in OUD, including a broad interferon response among glial cells and elevated DNA damage in neurons. DNA damage markers were identified in both the striatum of humans and monkeys, suggesting chronic opioid use associated with OUD leads to augmented DNA modifications (e.g., single- and double-stranded breaks, impaired DNA repair, and chromatin accessibility). Proinflammatory signaling, the involvement of microglia-dependent signaling in striatum, and alterations in synaptic signaling in OUD are consistent with previous bulk transcriptomics findings from post-mortem brains of individuals with OUD[2,5,9].

Many of the genes and pathways enriched in specific striatal cell types of individuals with OUD have been implicated in various processes related to cell stress and senescence, DNA damage, and inflammation. Opioids lead to persistent changes in neuronal activity and synaptic plasticity within the striatum. Prolonged or repeated alterations in neuronal activity combined with dysfunction within DNA repair pathways hampered by persistent cell stress (e.g., oxidative stress) may lead to the accumulation of DNA damage. Neurons may be particularly vulnerable to DNA damage[105], in that breaks in the DNA and subsequent DNA repair, allow neurons to respond rapidly at the transcriptional level to changes in activity[106,107]. DNA integrity is critical for neural cells to prevent insertions, deletions, or mutations that ultimately have consequences on cell health and function. Our results suggest elevated DNA damage in neurons of individuals with OUD, particularly within striatal interneurons.

Several subclasses of striatal interneurons are found in striatum[108], with different transcriptional and physiological properties. We separately clustered TH +, CCK +, PTHLH +, and SST+ interneurons in human striatum. The functionality of interneurons in the striatum is

diverse, with several subclasses modulating the excitability and inhibition of MSNs in response to various substances, including opioids[109]. Although we were unable to reference the overall elevation of DNA damage signature to a specific subset of interneurons, our findings suggest striatal interneurons may be particularly vulnerable to a loss of DNA integrity and repair mechanisms in response to changes in neuronal activity by opioids and related perturbations.

The DNA damage response in neurons can lead to a proinflammatory response in microglia and other glial cells[13,84]. In OUD, there were a relatively large number of genes that were upregulated specifically in microglia, enriched for genes involved in neuroinflammation. Microglial activation may propagate to other neural cell types, evident by an increase in interferon responses. Indeed, we found significantly upregulated interferon response in both neuronal and glial cells of individuals with OUD. Interferon responses may modulate opioid actions in the brain and could be involved in the emergence of symptoms of opioid withdrawal[110].

Other relevant pathways to OUD included oxidative stress, mitochondrial respiration, and neuroprotection. The coordinated alterations in the expression of genes involved in these pathways may reflect compensation to changes in neural activity impacted by opioid use. For example, glutamatergic hyperactivity at striatal MSNs[111] may cascade towards excitotoxicity and elevated DNA damage[112], both of which have been recently associated with OUD[113]. GRM5 was specifically downregulated in astrocytes in OUD. GRM5 is transiently expressed in astrocytes to detect and respond to extracellular glutamate, suggesting downregulation in OUD may be due to changes in glutamatergic activity in striatal neurons[114]. Alterations in other glutamatergic and GABAergic transcripts were also found in specific cell types in OUD (e.g., downregulation of *GRIA1* in astrocytes and upregulation of *GABRG2* in microglia). In parallel, enrichment of neurodegenerative-related and neuronal activity markers in certain striatal cells in OUD may be a result of high energetic demand on MSNs to maintain a hyperpolarized state, a possible mechanism of vulnerability of striatal MSNs proposed in Huntington's disease[115]. We also found OUD-associated changes in neuronal energetics with the downregulation of nicotinamide riboside kinase 1, *NMRK1*, and the basic helix-loop-helix family member e40, *BHLHE40* (DEC1), both of which are involved in regulating cellular metabolism, oxidative stress, and inflammation[81]. Together, the broad set of changes in genes and pathways associated with OUD were associated with various processes involved in cell stress.

Sex differences in the vulnerability to substance use, development of substance use disorders, and treatment outcomes, along with the biological response to drugs are evident from both preclinical and clinical studies[116–118]. Our findings suggest sex-specific alterations in the transcriptional response to opioids across striatal cell types. Based on the transcriptional patterns, we found a more pronounced inflammatory response associated with striatal microglia in females compared to males with OUD, suggesting microglia in the human brain also exhibit sex-specific responses to stress and substances[119,120]. In

addition, we identified the upregulation of *FKBP5* as a potential female-specific factor in several glial subtypes in OUD. *FKBP5* acts as a co-chaperone of the glucocorticoid receptor activated in response to stress, with major implications in the pathology of several psychiatric disorders and the impact of stress on substance withdrawal, craving, and relapse[121,122]. Further, the transcriptional alterations within microglia, oligodendrocytes, and astrocytes in female individuals with OUD were significantly enriched for *FKBP5* target genes identified in rodents administered opioids. Genes related to synaptic functions were also enriched in astrocytes of females, while enrichment was mostly in MSNs and interneurons in male individuals with OUD, suggesting sex-specific alterations of striatal neuronal and glial cell signaling in opioid addiction.

Several limitations of our resource stem from challenges of profiling single cell transcriptomes in postmortem human brains. While the nuclear transcriptome correlates on whole with the transcriptomes of other subcellular compartments, there are notable differences in mRNA trafficking within certain neural cell types and synapses. Furthermore, the nuclear transcriptome does not capture post-transcriptional regulation. Further, the sex-biased differential expression in OUD is likely influenced by cultural and behavioral factors that are confounded with biological sex. Lastly, while this resource will be useful to identify the consequences of OUD in the postmortem brain transcriptome, integration of our findings with other types of approaches from genetics to functional genomics in clinical cohorts to animal models of opioid addiction could contribute to a greater understanding of opioid addiction and aid in the realization of treatment strategies.

## Methods

### Human individuals

Postmortem human brain samples were obtained, following consent from the next of kin, during autopsies conducted by the Allegheny County Office of the Medical Examiner (Pittsburgh, PA). Consent was obtained from next-of-kin and procedures were approved by the University of Pittsburgh's committee for Oversight of Research and Clinical Training Involving Decedents and Institutional Review Board for Biomedical Research. An independent committee of clinicians made consensus, lifetime DSM-IV diagnoses for each individual using the results of an expanded psychological autopsy, including structured interviews with family members and review of medical records, as well as toxicological and neuropathological reports[123]. The same approach was used to confirm the absence of lifetime psychiatric and neurologic disorders in the unaffected comparison individuals. All procedures were approved by the University of Pittsburgh Committee for Oversight of Research and Clinical Training Involving Decedents and Institutional Review Board for Biomedical Research. Each individual meeting diagnostic criteria for OUD at the time of death ($n = 6$) was matched with an unaffected comparison individual ($n = 6$) for sex and as closely as possible for age and PMI (see Supplementary Data 1–S5). The duration of illness for each individual with OUD was at least four years prior to death.

For all individuals, the caudate and putamen were identified on fresh-frozen coronal tissue blocks using anatomical landmarks and tissue was collected via cryostat, using an approach that minimizes contamination from white matter and other striatal subregions and ensures RNA preservation. Fresh-frozen, right hemisphere coronal tissue blocks containing the body of the caudate and putamen, inclusive of plates 24–30 in the rostro-caudal axis, were included for analysis. The rostral face of each block was scored using a #11 scalpel blade while mounted in the cryostat. Sections were cut at 40 μm thickness, and each striatal subregion from an individual section was placed into its respective collection tube until a total of volume of ~50 mm$^3$ was collected from each region. This approach minimizes contamination from white matter and other striatal subregions and ensures RNA

preservation. The medial-lateral border between the caudate and putamen was scored to exclude the internal capsule. The lateral border of the putamen was defined by the external capsule. The ventral border of the caudate and putamen was defined by the anterior thalamic radiation of the internal capsule or the anterior commissure.

### Isolation of nuclei from human postmortem brain tissue and library preparation

Nuclei were isolated from 24 biospecimens of frozen human postmortem brain tissue (12 individuals (Unaffected/OUD) x 2 brain regions = 24 samples). Samples weighing 10–15 mg were homogenized using ~10 strokes per glass pestle in 7 mL glass douncers with 5 mL nuclei isolation medium containing DAPI. Homogenate was filtered using a 40um mesh strainer (Fisher Scientific #48680). Nuclei were sorted for DAPI fluorescence using a BD FACS Aria at the Boston University Flow Cytometry Core. Approximately 100,000 nuclei were sorted into 7ul of 0.04% bovine serum albumin (Millipore Sigma #126615) in phosphate buffered saline (ThermoFisher #10010031). Nuclei were counted using a hemocytometer and assessed for concentration and debris. 7000 nuclei were targeted per sample except for one sample with lower concentration where 5000 nuclei were targeted. The 10x Chromium process was performed and next generation sequencing libraries were prepared using the 10x genomics single cell 3' gene expression dual index kit.

Libraries were sequenced at the Boston University Single Cell Sequencing Core. The pool of snRNA-seq libraries were sequenced on 7 Next-seq P3 flow cells with an intermediate re-pooling scheme to optimize for 50–80% sequencing saturation, > 8000 average UMI per cell. Between sequencing runs, we preliminarily aligned the sequencing reads as outlined below to assess quality per sample and estimate the number of viable nuclei and sample complexity. We identified two samples, C-13291 and C-612, to have low QC metrics due to wetting failures, mean UMI per cell <1000 and estimated # cells >50,000. These samples were excluded from subsequent re-pooling and further analyses (Supplementary Data 1–S1: tab STARsolo QC).

### Nonhuman primate subjects

Eight adult rhesus macaque monkeys (*macaca mulatta*) weighing between 6.0 and 13.0 kg served as subjects in the present study. Subjects lived individually in stainless-steel enclosures under a 12 h light-dark cycle with side and front visual access to other conspecifics. All subjects had continuous access to water and were fed a diet of High Protein Monkey chow (Purina Mills International, Brentwood, MO), fresh fruit, and vegetables. Environmental enrichment (mirrors, toys, foraging boards, music, etc.) was also provided daily. Subject health and well-being were monitored daily by trained technical and veterinary staff. Animal husbandry and research was conducted in accordance with the guidelines provided by the Institute of Laboratory Animal Resources as adopted and promulgated by the U.S. National Institutes of Health. The facility is licensed by the U.S. Department of Agriculture and all experimental protocols were approved by the Institutional Animal Care and Use Committee at McLean Hospital.

### Chronic morphine dosing in nonhuman primate subjects

Four subjects (3 male, 1 female; mean age: 12 years, range: 9–18 years) received daily morphine treatment for 5 months (see below); four additional subjects, matched for age, weight, and sex, served as experimental controls (3 male, 1 female; mean age: 14 years, range: 11–21 years). Control subjects had a history of nicotine or cocaine exposure but were drug free for ~1 year prior to tissue collection. The metadata for nonhuman primate subjects are listed in Supplemental Supplementary Data 1–S26.

Each morphine-treated subject was trained to come to the front of the enclosure for twice daily intramuscular (IM) injections (< 0.5 cc) of morphine sulfate (NIDA Drug Supply Program) dissolved in 0.9%

saline. Injections were administered at 09:30 and 17:00 h. A gradual dosing escalation method (0.5 log unit increase every 3 days) was used to achieve the terminal dosage of 9 mg/kg/day (i.e., 4.5 mg/kg, BID). This dosage was selected to produce moderate opioid dependence[124,125]. All subjects received approximately 1500 mg of morphine over the 5–6-month period of chronic dosing. On the last day and approximately 3 h following the last IM morphine injection, each subject received an IM injection of ketamine (10 mg/kg) followed by 5.0 ml IV of a pentobarbital-based euthanasia solution (Beuthanasia-D).

## Rhesus brain tissue preparation

After sacrifice, brains were rapidly dissected into slabs and frozen on metal plates in liquid nitrogen vapor. Time between animal sacrifice and tissue freezing was between 1 and 2 h for all animals. Brains were kept stored at −80 °C until punching, when they were punched on a microtome with guidance from a macaque anatomist (Dr. S. Haber). Punches were stored at −80C until the day of nuclear isolation and encapsulation.

## Single nucleus RNA sequencing in rhesus brain tissue

Nuclei were isolated as in previous studies[126], with minor modifications. Punches were placed into buffer HB (0.25 M sucrose, 25 mM KCl, 5 mM MgCl$_2$, 20 mM Tricine-KOH pH 7.8, 1 mM DTT, 0.15 mM spermine, 0.5 mM spermidine, protease inhibitors) and placed in a Dounce homogenizer for 10 strokes each with loose and tight pestles. A 5% IGEPAL solution was added to a final concentration of 0.3% followed by five additional dounce strokes, then the lysate was filtered through a 40 μm strainer. Nuclei were mixed with an equal volume of 50% iodixanol and then layered on top of an iodixanol gradient of 40% and 30% layers in a 2 mL dolphin microcentrifuge tube. Nuclei were spun by centrifuging at 10,000 x g for 4 min at 4 °C and then collected by aspiration at the interface of the 30% and 40% iodixanol layers. Nuclear concentration and prep quality were ascertained by loading on a hemocytometer and were diluted to a concentration of 80–100 K and 15% iodixanol with Buffer HB prior to loading on InDrops v3 platform. Single-nuclei suspensions were encapsulated into droplets, lysed, and the RNA within each droplet was reverse-transcribed using a unique nucleotide barcode[127]. Approximately 6000 nuclei, in two batches of 3000 nuclei each were processed per library and sequenced on Illumina NovaSeq S2 chips (at a density of approximately 20,000 reads/nucleus).

## Single nuclei RNAseq data processing

We aligned single nuclei RNA-seq (snRNA-seq) reads to the human genome (GRCh38.p13) or rhesus macaque genome (rheMac10) for each output with the turn-key single-cell transcriptomics method STARsolo, which is folds faster than the CellRanger pipeline and equally accurate (v2.7.9a)[128]. For the macaque samples, we used a set of gene annotations by mapping the human gene annotations to the rheMac10 genome using the liftoff tool[129]. These alternate rheMac10 gene annotations are deposited to Carnegie Mellon University's Kilthub repository resource (https://kilthub.cmu.edu/articles/dataset/Alternate_gene_annotations_for_rat_macaque_and_marmoset_for_single_cell_RNA_and_ATAC_analyses/21176401).We chose parameters for the STARsolo UMI quantification to closely replicated the 10X CellRanger pipeline v6 and use the filtered genome and gene annotation available from 10X Genomics (https://support.10xgenomics.com/single-cell-gene-expression/software/downloads/latest, Human reference 2020-A). We ran STARsolo to allow for pre-mRNA gene counts as well as exonic counts for nuclear RNA and to separately count introns and exons for RNA velocity analyses, (--soloFeatures GeneFull Velocyto). We used the following parameters to correct cell barcodes, de-duplicate transcripts by their unique molecular identifier (UMI), assign UMI counts to genes, and pre-filter cells that are likely empty droplets

(--soloType Droplet --soloCBmatchWLtype 1MM --soloCellFilter EmptyDrops_CR --soloMultiMappers EM --soloUMIdedup 1MM_CR). We pre-processed the UMI gene x cell count matrix to reduce inherent biases in the technology. We identified likely ambient RNA contamination with SoupX[130], empty droplets with DropletQC[131], doublets with scds[132], and damaged nuclei with miQC[133]. For each of these analyses, each sample (GEM well) was analyzed separate from each other. We ran SoupX to estimate the fraction of ambient RNA from both raw and unfiltered UMI count matrices from STARsolo and perform ambient RNA removal aware of the cell clusters in the filtered matrix. For just the SoupX analyses, we clustered the cells with Seurat v4[134] with FindClusters (algorithm = 2, resolution = 0.5). For DropletQC, we used the intronic and exonic UMI counts per cell per gene from STARsolo to get the fraction of intronic UMI per cell (referred to as the nuclear fraction). We identified empty droplets with default DropletQC parameters (nf_rescue = 0.50, umi_rescue = 1000). We identified droplets with scds's hybrid algorithm using the function cxds_bcds_hybrid to estimate doublet scores and called doublets on cells with scds.hybrid_score > 1.0. We identified damaged cells with high percentage of mitochondrial UMI counts using only miQC which uses a Bayesian EM algorithm to learn the relationship between mitochondrial UMI counts and number of captured genes. We used the posterior probability cutoff of 0.75 to call damaged cells by miQC.

To combine cells together across samples, we normalized the UMI counts with the variance-stabilizing SCTransform and glmGamPoi on each sample[135,136] and jointly embedded cells across samples with reciprocal PCA integration[134], as outlined in https://satijalab.org/seurat/articles/integration_rpca.html. In this joint embedding, we over-clustered the dataset with FindClusters(algorithm = 2, resolution = 1)and removed any cluster with more than 10% of cells flagged by miQC, scds, or DropletQC as low-quality biased-clusters in the data.

## Labeling striatum cells with a macaque snRNA reference dataset

We annotated our cells from the human or monkey striatum to a recently published high-resolution snRNA-seq reference dataset of the non-human primate striatum[34] using Seurat v4. We downloaded the monkey snRNA-seq processed, annotated gene UMI counts for all cells and MSNs from GSE167920. He, Kleyman et al. had aligned the snRNA-seq reads to the rheMac10 genome using the GRCh38 gene annotation liftover to rheMac10, so the gene-wise labels represent the UMI counts on the rheMac10 genome most orthologous to human. For both full nuclei and MSN subset datasets, we re-processed the macaque cells with SCTransform, glmGamPoi, and reciprocal PCA with default parameters as above to enable label transfer using the most recent integration algorithms in Seurat.

To transfer cell annotations from the reference macaque striatum dataset to the human or macaque striatum cells, we performed two label transfers at increasing resolutions: one with all cells and another with just MSNs. As He, Kleyman et al. described the differences between transcriptionally and anatomically distinct MSN subtypes are subtle, so we split the annotations into two steps to optimally annotate the cells. The first label transfers the cell classes (Oligodendrocytes, MSNs, Interneurons, etc.) from the macaque to the human dataset with the Seurat functions FindTransferAnchors (reduction = 'rpca') and TransferData. Next, we identified cells or cell clusters that were labeled as MSNs and transferred MSN subtype labels (D1.Striosome, D2.Striomsome, etc.) from the macaque to human datasets. We filtered out cells where the cell class or cell subtype labels have max prediction scores less than 0.5 as these tend to represent noisy predictions due to low quality cells from either dataset. We confirmed accurate label transfer at the cell class and cell subtype levels with published marker genes and similar proportions across individuals and samples.

Even with the robust cutoffs that we applied to this dataset to remove likely low quality or doublet cells; we found a residual subset of the data that contain these cells. Upon clustering, doublet cells tended

to project into the UMAP space as long streaks between two well-defined cell types. Low-quality cell types would project into the UMAP space as amorphous cell types without clear boundaries. Using these embedding features, we selected these clusters with Seurat's Find-Clusters(resolution = 1) function, confirmed that they have the indicative QC metrics, and removed them from analyses.

## Annotating interneurons with mouse marker genes

To demonstrate the high resolution of our datasets, we annotated the striatal interneuron using previously characterized mouse markers of these subtypes[108]. We sub-clustered the interneurons labeled by the macaque dataset and annotated them manually as interneuron subtypes best labeled by the marker genes *TH*, *PTHLH*, *SST*, or *CCK*. The previously published macaque snRNA-seq dataset had too few of these interneurons sampled from $N = 2$ monkeys, so these subtypes were under-represented to be sub-clustered. In this study, we sampled more broadly from $N = 12$ individuals (human) or $N = 8$ individuals (rhesus macaque). While these interneuron subtypes are clearly distinct in this dataset and in the neural circuits, they still are under-represented to power certain downstream. For these analyses, we analyze these cells together as "Interneurons".

## Differential gene and cellular state expression analysis in humans

To investigate the gene expression differences in OUD and unaffected individuals, we used the pseudo-bulk aggregation of gene expression profiles. Many have shown that pseudo-bulk-based case-control differential expression analyses robustly detect gene-level differences with lower false discovery due to repeated measures from single cells of the same individual[137–139]. The raw UMI counts were added together from the same individual, brain region, and cell type to create the pseudo bulk profiles. We aggregate the interneuron subtypes together as "Interneurons". We filtered out pseudo-bulk profiles aggregated from more than 20 cells. We filtered out mitochondrial, ribosomal, and low-expressing genes with less than 5 average UMI counts. We retained 20,203 genes and 210 pseudobulk profiles to apply the voom-limma method[140] for differential gene expression analyses and the sva method to construct surrogate variables to identify un-modeled sources of transcriptomic variation. These statistical methods together addressed several challenges in analyzing cell type differential expression across multiple axes of meaningful biological variation: 1) accounting for correlated pseudo-bulk samples shared across individuals with the duplicateCorrelation (block = Subject.ID), 2) estimating quality weights for adjusting for cell type proportions with the function voomWithQualityWeights, and 3) calculating un-modeled variation in high-dimensional single cell data with the sva()function. To plot the gene expression profiles, we used the normalized counts per million (CPM) of each pseudobulk profile corrected for the batch effects and surrogate variables unrelated to the OUD diagnosis using the cleaning function (https://github.com/LieberInstitute/jaffelab). For heatmap visualizations of the gene expression profiles, we also z-normalized the corrected expression profile of each gene grouped by cell type, since gene expression is highly cell-type specific.

We calculated the differential expression in OUD vs. unaffected comparison individuals for two sets of hypotheses:

1. What is the differential expression in each cell type averaged across brain regions and sex?
2. What is the differential expression within the subset of female or male individuals?
3. What is the differential expression interaction between sex and OUD?

To achieve these comparisons, we used one linear model with a nested variable capturing the interaction between cell type, OUD diagnosis, Sex, and Region, Celltype_Dx_Sex_Region. The nested variable allowed for contrasting subsets of the data to calculate differential expression for each family of hypotheses at different cell type resolutions:

$$\text{Gene\_expression} \sim 0 + \text{Celltype\_Dx\_Sex\_Region} + \text{Age} + \text{PMI} + \text{RIN} + \text{GDR} + 23\text{SVs},$$

(GDR: gene detection rate, SV: surrogate variables). We estimated the effect of OUD vs. unaffected individuals across each annotated cell type, across neuronal subtypes (Neuron), across glial types (Glia), and across all cell types (All). For the first set of hypotheses at the annotated cell type level, we used the simple contrast CelltypeA_OUD_Sex?_Region? - CelltypeA_UC_Sex?_Region? with samples from both Sex and Regions to obtain the average effect across those variables (?, the wildcard placeholder for F, M, Caudate, or Putamen). For the second set hypotheses, we only included individuals within each Sex to obtain the effect of OUD vs. unaffected within each sex subset. For the third set of hypotheses, we use the following contrast: (CelltypeA_OUD_SexF_Region? - CelltypeA_UC_SexF_Region?) - (CelltypeA_OUD_SexM_Region? - CelltypeA_UC_SexM_Region?).

We calculated the differential expression in chronic morphine exposure vs. non-morphine treat rhesus macaque subjects. We used one linear model with a nested variable capturing the interaction between cell type, morphine treatment and matching stats.

$$\text{Gene\_expression} \sim 0 + \text{Celltype\_Tx} + \text{Pair} + \text{SVs}.$$

We estimated the effect of morphine treatment vs. no-morphine individuals across each annotated cell type, across neuronal subtypes (Neuron), across glial types (Glia), and across all cell types (All). For the first set of hypotheses at the annotated cell type level, we used the simple contrast CelltypeA_morphine - CelltypeA_control.

## Gene set enrichment analyses

We identified pathways that are differentially altered in OUD using gene set enrichment analyses with the molecular signatures database[55] and the fgsea and msigdbr R-packages[141]. We included the hallmark gene pathways, curated gene sets from BioCarta, KEGG, Canonical Pathways, Reactome, and WikiPathways, and ontology gene sets. We also downloaded and included curated pathways from SynGO in enrichment analyses[142]. We report the full list of enriched pathways corrected for multiple hypotheses within each set of hypotheses in Supplementary Data 1–S7. We clustered redundant/related pathways using custom R-scripts and the igraph R-package creating networks of pathways connected by overlapping genes[143,144] (igraph.org). We visualized the clustered pathways using igraph functions and report both clustered and singleton pathways alongside each figure in Supplementar Data 1–S7, S9, and S11. For the interaction score for calculating pathway-enrichments between female- and male-biased pathway analyses, we used the log2(fold-change) and the p-values from the OUD v. UC effect within differential analysis of males or females to calculate an interaction score: $\text{sign}(\log_2 FC_F) * (-\log_{10}(\text{p-value}_F)) - \text{sign}(\log_2 FC_M) * (-\log_{10}(\text{p-value}_M))$. We overlapped enriched pathways using this calculation vs. the pathways enriched in the standard OUD v. UC in females or males to further support the interpretation of female- or male-biased pathways.

To intersect the genes found to be differentially expressed in OUD within females and males, we also compared to differentially expressed genes identified by microarray studies of striatum of male mice exposed to drugs[104]. Piechota et al. identified from hierarchical clustering 2 main gene signatures that became pronounced time-lapsed since drug exposure, A and B, with subsets of the second B1, B2, and B3. We identified the human orthologs of these gene sets and performed GSEA as above on the differentially expressed genes identified across the full cohort, on the subset of female and male individuals, or the

caudate and putamen samples. We report the main findings in the text and the full set of enrichments in Supplementary Data I–S16.

### Transcription factor-gene regulatory network analyses

The single nuclei RNA-seq dataset in this study provides a resource to generate gene pathways. To discover pathways from our single cell RNA-seq, we applied the pySCENIC Protocol to build transcription factor (TF)- gene regulatory networks[47,145]. To potentially capture individual-specific TF-gene regulatory relationships, we ran the GRNBoost2 with multiprocessing to infer TF-gene relationships from the pySCENIC package on each individual separately[145,146]. To speed up the GRNBoost2 run time, and reduce bias by cell type proportion, we down-sampled oligodendrocytes (~60% of all cells) to the next most prevalent cell type, Astrocytes (~8%), reducing the GRNBoost2 run time by 50%. Furthermore, we subset the ~20k genes that are expressed and analyzed in the pseudobulk differential expression analysis further reducing GRNBoost2 runtime by 30%. Since GRNBoost2 is a stochastic method, we ran this step 3–4 times for each individual and selected the TF-gene relationships that were reproducible > 80% across runs. Similarly, to aggregate the TF-gene relationships across individuals, we aggregated TF-gene relationships that were detected in 80% of 12 individuals. Across these aggregation steps, we averaged the importance scores for selected TF-gene relationships. We selected likely direct TF-gene relationships by filtering relationships where the gene's promoter contain the TF's binding motifs using the cisTarget method implement in pySCENIC package using the pre-built TF-motif promoter binding databases for human accessed from (https://resources.aertslab.org/cistarget/databases/ in January 2023)[47]. We report the list of discovered TF-gene relationships in the Supplementary Data 1–S17.

### Analyses of cellular activation of gene sets

We calculated the level of activity of each collection of gene sets with the AUCell method[47] on the list of TF-gene regulatory networks or custom gene sets (described below). For the TF-gene relationships identified above, we calculated the differentially active TF-gene modules between unaffected and OUD individuals by creating the pseudobulk average of each TF-module by individual, brain region, and cell type. We performed linear regressions to assess the effect of OUD diagnosis accounting for covariates RIN, Age, Sex, Region, and number of aggregated cells. We corrected multiple testing of the OUD effect on TF-gene module activity across all cell types and TF modules and reported the full cell type by sample linear regression are reported in Supplementary Data 1–S8. We similarly performed a similar pseudobulk average at the biological sample level aggregating over individual and brain regions. We visualized the significant TF-gene module activity differences set at FDR < 0.05 in glia and neurons and of select modules with the igraph R-package (igraph.org).

In addition to the TF-gene relationships, we assessed the state of DNA damage in OUD using two published gene signatures of DNA damage[13]. The neuronal DNA damage signatures were collected by Welch et al. from fluorescence activated sorting of mouse brain nuclei gated for NeuN + :gH2AX+ nuclei. Using the human orthologs of these mouse genes, we scored our single nuclei RNA-seq datasets with the AUCell method to study the differences in DNA damage gene signatures between unaffected and OUD individuals or no-morphine or chronic morphine treated rhesus macaques. We applied the same pseudobulk average of DNA damage scores and reported the linear regression interaction between OUD and cell type in DNA damage signatures in Supplementary Data 1–S10. We similarly applied the same pseudobulk average of DNA damage scores and report the linear regression interaction between chronic morphine exposure and cell type regressing out the match pair in DNA damage signatures.

### High-dimensional weighted gene co-expression network analysis (hdWGCNA)

hdWGCNA was used to further analyze gene co-expression networks across striatal cell types[147]. Each of 6 cell types of interest (D1-Striosome, D2-Striosome, D1-Matrix, D2-Matrix, D1-D2 hybrid and microglia) was used as input. Gene expression data was SCT transformed. Each dataset was then collapsed into a single metacell. Softpower was selected for each of the metacell to construct gene co-expression networks. The gene modules were identified using unsupervised clustering via the Dynamic TreeCut algorithm with default settings. For each module, Top 100 hub genes for each module were identified. Metascape (metascape.org)[148] was used for pathway analysis on modules from co-expression gene networks by each cell type (Fig. S11; Data 1–S18). Eigengene of identified modules was correlated with "traits" (in our case, OUD, Sex, Race and RIN, number of features, number of RNA count and percent of mitochondrial RNA) using Pearson correlation. Modules with significant correlation ($p < 0.05$) with OUD were used for downstream analysis (Data 1–S21, S22), especially, modules significant with OUD (Figs. S9, 10, 11; Data 1–S18, S19) were prioritized. For cell types with no module uniquely correlated with OUD, hub genes from all OUD trait correlated modules were used as input for pathway analysis (Data 1–S23–S25).

### Statistical analyses

We corrected for multiple hypotheses testing in this study using the false discovery rate (FDR) less than alpha = 0.05. In different expression analyses, we perform FDR correction across all cell types tested. Similarly for gene set enrichment analysis and AUCell pseudobulk analyses, we perform FDR correction across all pathways and differentially expressed genes across cell types. We computed the FDR correction with the swfdr R package which increases power by leveraging Storey's q-value and modeling the relationship of null P-values and independent variables such as the average logCPM or the number of genes in a pathway[149,150]. In differential gene expression analyses, we calculated the FDR within each cell type comparison (adj.P.Val.Within) and between all cell type comparisons (adj.P.Val.Between). We report the more conservative correction between cell types in the main text and figures, adj.P.Val.Between. Measured metrics (gene expression counts, quality control metrics) or derived metrics (AUCell gene activity scores) are repeated measurements of the same cohort of human individuals or rhesus macaque subjects. Metrics that are differentially expressed across multiple cell type conditions are reported and summarized at the largest significant adjusted P-value and lowest effect size magnitude. Genes or metrics that are only significantly differentially expressed across one cell type are reported with exact adjusted P-values and effect sizes rounded to 2 significant digits. Exact P-values and effect sizes are reported in the supplementary data. All statistical analyses performed use two-sided test statistics.

For human samples, linear regressions of differential AUCell modules, or other quality control metrics at the pseudobulk analyses were performed across biospecimens ($N = 22$ biospecimens) controlling for relevant covariates including brain region extracting the interaction between OUD diagnosis with cell type. The general linear model for this linear regression for sample level is

$$\sim Dx + Age + PMI + RIN + Sex \text{ and for cell type level is}$$
$$\sim Dx : Celltype + Celltype + Age + PMI + RIN$$

where we extracted the interaction term to determine OUD diagnosis-specific changes in a cell type. For rhesus macaque samples, linear regressions of differential expression and differential AUCell modules were performed across biospecimens ($N = 8$ biospecimens) controlling for matched pair grouping variable, which accounts for sex, age, and

weight. The general model for this linear regression at the individual level is

$$Value \sim Tx + Pair \text{ and for cell type level is } Value$$
$$\sim Celltype : Tx + Celltype + Pair.$$

## Reporting summary

Further information on research design is available in the Nature Portfolio Reporting Summary linked to this article.

## Data availability

The raw sequencing reads and annotated Seurat objects for both human and rhesus macaque studies are uploaded to GEO under SuperSeries accession number GSE233279. A browsable webportal of the human dorsal striatum single nuclei transcriptomes are on the CZ CellxGene Discovery webportal (https://cellxgene.cziscience.com/collections/cec4ef8e-1e70-49a2-ae43-1e6bf1fd5978). Source data are provided in unified file for manuscript.

## Code availability

Single nuclei RNA-seq data processing and downstream analyses of the human dorsal striatum in this paper are collected in the github repository (https://github.com/pfenninglab/Logan_Striatum_snRNA-seq) and deposited at https://doi.org/10.5281/zenodo.10433681. Single nuclei RNA-seq processing and downstream analyses of the rhesus macaque nucleus accumbens in this paper are collected in the github repository (https://github.com/pfenninglab/McLean_chronic_opioid_monkey_snRNA-seq) and deposited at https://doi.org/10.5281/zenodo.10433683. Source data are provided in unified file for manuscript.

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

## Acknowledgements

Thank you to the staff and technicians who work diligently as part of the Brain Tissue Donation Program at the University of Pittsburgh. Postmortem human brain tissue was provided by the University of Pittsburgh Brain Tissue Donation Program and the National Institutes of Health NeuroBioBank at the University of Pittsburgh. The research reported in this article was supported by: Research was supported by NIH HEAL Initiative under National Heart, Lung, and Blood Institute (R01HL150432, R.W.L.); and National Institute on Drug Abuse (R01DA051390, M.L.S. and R.W.L.; DP1DA046585, A.R.P.; F30DA053020, B.N.P.; R01DA047130, S.J.K.).

## Author contributions

R.W.L. conceptualized and managed the project. R.W.L. and A.R.P. provided supervision for data analyses and interpretation. B.N.P., A.R.P., R.W.L., S.J.K., and M.L.S. obtained funding for the project. B.N.P., M.H.R., J.F.G., D.A.L., Z.F., M.L.W., A.R.P., and R.W.L. designed and implemented the project. M.H.R., M.K.F., A.E.T., S.J.R., and Y.A. conducted human tissue homogenization, single nuclei extraction, and single nuclei library preparations. R.J.F., S.J.K., J.B., S.N.H., K.M.M., and K.J.R. conceptualized and managed the rhesus macaque project, including tissue homogenization, single nuclei extraction, and single nuclei library preparations. B.N.P. and C.F. completed the computational analyses and B.N.P., M.H.R., Q.S., and R.W.L. provided software, visualization and

validation. B.N.P., X.X., Q.S., G.C.T., A.R.P., and R.W.L. designed and interpreted analyses and outcomes. R.W.L., J.G., D.A.L., and ARP provided the tissue resources and computational infrastructure for analyses. B.N.P., M.H.R., M.K.F., A.R.P., and R.W.L. wrote the original manuscript with figures. All authors participated in reviewing and editing the manuscript for publication.

## Competing interests

The authors declare no competing interests.

## Additional information

[1]Computational Biology Department, Carnegie Mellon University, Pittsburgh, PA 15213, USA. [2]Neuroscience Institute, Carnegie Mellon University, Pittsburgh, PA 15213, USA. [3]Medical Scientist Training Program, University of Pittsburgh School of Medicine, Pittsburgh, PA 15213, USA. [4]Department of Pharmacology, Physiology & Biophysics, Boston University School of Medicine, Boston, MA 02118, USA. [5]Whitaker Cardiovascular Institute, Boston University School of Medicine, Boston, MA 02118, USA. [6]Department of Biostatistics, University of Pittsburgh, Pittsburgh, PA 15213, USA. [7]Department of Psychiatry, University of Massachusetts Chan Medical School, Worcester, MA 01605, USA. [8]Department of Psychiatry, Harvard Medical School, Boston, MA 02115, USA. [9]Division of Depression and Anxiety, McLean Hospital, Department of Psychiatry, Harvard Medical School, Belmont, MA 02478, USA. [10]Behavioral Biology Program, McLean Hospital, Belmont, MA 02478, USA. [11]Department of Pharmacology and Physiology, University of Rochester, School of Medicine, Rochester, NY 14642, USA. [12]Basic Neuroscience Division, Department of Psychiatry, Harvard Medical School, McLean Hospital, Belmont, MA 02478, USA. [13]Center for Systems Neuroscience, Boston University, Boston, MA 02118, USA. [14]Graduate Program for Neuroscience, Boston University, Boston, MA 02118, USA. [15]Department of Psychiatry, University of Pittsburgh School of Medicine, Pittsburgh, PA 15219, USA. [16]Department of Cell Biology, University of Pittsburgh School of Medicine, Pittsburgh, PA 15219, USA. [17]Department of Pathology and Laboratory Medicine, Boston University School of Medicine, Boston, MA 02118, USA. [18]Department of Neurobiology, University of Massachusetts Chan Medical School, Worcester, MA 01605, USA. ✉e-mail: apfenning@cmu.edu; ryan.logan@umassmed.edu

