## [Peer Review File · Nature Communications]

Single nuclei transcriptomics in human and non-human primate striatum in opioid use disorderReviewers' comments:

Reviewer #1 (Remarks to the Author):

I appreciate the authors taking the effort to respond to many of my questions. While some issues were either ignored (comments about data presentation) or not effectively addressed (data analysis/QC), the dataset could still be useful as a resource. In particular, I was pleased by the reduction in focus on neurodegeneration.

The remaining focus on sex differences are still either underpowered or under explored with respect to the analysis. For example, no statistical test was performed to support the conclusion of differential impact of sex on DEGs in neurons compared to glia. Such comparisons could be complicated by a number of factors such as the number of cell types, heterogeneity, etc. that do not seem to be addressed. In addition, the pathway enrichment in females with OUD are difficult to discern from the figure. I still think it would be beneficial to provide the reader with the differentially expressed genes related to sex differences in OUD rather than pathways.

With respect to pathway analysis, the author response about not ranking pathways and not comparing pathway enrichment scores puzzles me. Such analysis would be valuable to the reader.

The authors also did not provide the UMI and expressed gene number by cluster and/or sample. Feature plots and expression of individual genes such as FOXP2 are not sufficient as either quality control or cluster annotation.

Reviewer #2 (Remarks to the Author):

Overall, the authors have addressed almost all of my concerns from prior review (at a different Nature title). The revised manuscript contains a number of significant text changes and additional analyses (e.g., Fig. S1, S2, S3, and S6). The results are noteworthy and report a very useful resource for the field. While I am supportive of the manuscript in its current written form (and based on the response to reviewers and the main text), it appears that upload/transfer issues may have resulted in the lack of revised main figures with this document. A couple key issues are outlined below:

1. The authors state in the response to reviewers that they have removed the original Figure 6, and the new manuscript text includes a figure caption for a new Figure 6 with different content. However, the old Figure 6 (containing a summary illustration of all findings, titled "Parallel cellular and molecular pathways in neurodegeneration and opioid use disorder") is still included in the paper.
2. Similarly, despite noted changes in the response to reviewers, I could not find any changes to the main figures since the original submission - perhaps the original figures were accidentally transferred over?
3. The review files appear to contain two versions of the supplemental materials - one is updated with new content and another is the original version (with no changes).
4. The response to reviewers notes that comparison of the relative proportions of all cell types across biosamples is now included in Figure 1, but I could not find this information - again suggesting that new main figures were not included.
5. The response to reviewers notes that PMID 30257220 has been included in the reference list, however I could not find this citation in the manuscript.

Overall, these issues make it difficult to fully evaluate the manuscript in its present form, although I think - from reading the response to reviewers - that it is likely this simply resulted from an upload/transfer error. Nevertheless, this issue should be corrected before I would be able to evaluate the full scope of the revisions.

Reviewer 1:

I appreciate the authors taking the effort to respond to many of my questions. While some issues were either ignored (comments about data presentation) or not effectively addressed (data analysis/QC), the dataset could still be useful as a resource. In particular, I was pleased by the reduction in focus on neurodegeneration.

We thank the reviewer for their constructive feedback to improve our manuscript. Our hope is we have addressed the remaining concerns of the reviewer. We appreciate the reviewer's support for our manuscript.

The remaining focus on sex differences are still either underpowered or under explored with respect to the analysis. For example, no statistical test was performed to support the conclusion of differential impact of sex on DEGs in neurons compared to glia. Such comparisons could be complicated by a number of factors such as the number of cell types, heterogeneity, etc. that do not seem to be addressed. In addition, the pathway enrichment in females with OUD are difficult to discern from the figure. I still think it would be beneficial to provide the reader with the differentially expressed genes related to sex differences in OUD rather than pathways.

We appreciate the reviewer's valuable input regarding our sex-specific findings. In response to this feedback, we have made several adjustments to our analysis and manuscript. We highlighted our previous analyses of differentially expressed genes (DEGs) comparing unaffected to OUD subjects within females or within males. DEGs are now color coded in Supplemental Table 12. These changes are aimed at enhancing the visibility of sex-specific DEGs, allowing viewers to identify significant DEGs in females, males, or both. We added a legend tab to Supplemental Table 12.

*We incorporated the reviewer's suggestion to provide information on DEGs by sex in Figure 6. Bar chart equivalents of Venn diagrams are included in **Figure 6A** to illustrate the number of significant DEGs by cell type in females, males, or both sexes. Notably, these figures depict that on average, there are more genes that are significant in females but not males in glial cell types (female only glia, 360 ± 60 ; male only glia, 212 ± 67). Similarly, female-only significant genes surpass male-only genes in neurons (female only neurons, 91 ± 9.1 ; male only neurons, 66 ± 7.9). This novel insight was brought to our attention by the reviewer, and we are genuinely grateful for the suggestion to simplify the communication of our findings.*

*To directly assess sex-specific differences, we calculated the interaction between sex and diagnosis, identifying DEGs specific to males and females in subjects with OUD. Our analysis revealed significantly more DEGs in glia relative to neurons (291 vs. 97) with sex-specific changes in OUD ($FDR < 0.05$). Notably, when we focused on the genes with interactions and identify which sex has the more significant effect (measured by $-\log_{10}(p\text{-value}_{\text{SexF}})$ vs $-\log_{10}(p\text{-value}_{\text{SexM}})$), we found more significant DEGs in females with OUD than male subjects with OUD (**Figure S13**). Results were added to Supplementary Table 12 and integrated into the main text.*

*We lastly identified sex-specific DEGs within diagnosis to determine whether biological sex in unaffected subjects contributed to the interaction of sex by OUD. We generated four-way Venn diagrams to assess overlaps of each of the sub-groupings (**Figure S14**). We found sex-specific DEGs in OUD were largely different from unaffected subjects, suggesting gene alterations in females and females were specific to OUD. Sex-specific gene alterations in OUD were similar gene changes in mice following opioid administration, identifying *Fkbp5* as a key, sex-specific factor in opioid addiction.*

With respect to pathway analysis, the author's response about not ranking pathways and not comparing pathway enrichment scores puzzles me. Such analysis would be valuable to the reader.

We appreciate the reviewer's feedback regarding our manuscript. We acknowledge the potential value of the suggested analysis, provided it is statistically feasible. As mentioned in our previous response, it's important to note that the NES value for any given pathway can be influenced by the number of factors that confound comparison between cell types and pathways. In the plot below, we show the absolute NES by $-\log_{10}(p\text{-value})$ along with the size of each pathway. We find that larger pathways (more enriched genes, larger circle) have a bias towards lower absolute NES. Similarly, the stronger significance ($-\log_{10}(p\text{-value})$) is positively correlated with absolute NES. This trend is present across all cell types. Notably, the canonical MSNs, interneurons, and endothelial cells have more enriched pathways than other cell types. When we inspect the pathways supplemental table, this is largely due to

overrepresentation of certain molecular signaling pathways selectively studied and reported such as VEGGF signaling and oxidative phosphorylation.

Additionally, due to the intricate interconnections among gene pathways, it can be challenging to disentangle the enrichment of one pathway from another. For instance, in **Table S9**, we observe enrichment in two pathways: "Electron transport chain: OXPHOS system in mitochondria" (NES = 2.4, # genes = 87, FDR = 0.0000132) and "Respiratory electron transport, ATP synthesis by chemiosmotic coupling, and heat production by uncoupling proteins" (NES = 2.3, # genes = 112, FDR = 0.0000138). To the best of our knowledge, there is no established algorithm or statistical method that definitively distinguishes which of these two pathways is more or less enriched in a manner that would alter the fundamental interpretation that both pathways are enriched. Due to the biases related to the NES, we interpret pathways by the strength of association, corrected for multiple hypotheses (FDR), which has a well understood definition, and the sign of the NES to denote up- or down-regulation. To assist readers, we provide the numerical results of these pathways in the supplemental tables. These tables offer the NES values, FDR, cell type, and enriched genes, allowing experts in these pathways to make their own judgments regarding the relative enrichment of pathways.

Given that existing pathway analyses often do not adequately address the interrelated nature of gene pathways, we conducted additional pathway clustering based on shared, significant genes to identify distinct biological processes that are affected (Methods). We believe that presenting distinct clusters of biological pathways makes our findings more interpretable and actionable for the broader scientific community. However, if the reviewer can provide a specific algorithm, statistical method, or a reference with detailed instructions, we are open to exploring an analysis that can address the mentioned statistical concerns.

The authors also did not provide the UMI and expressed gene number by cluster and/or sample. Feature plots and expression of individual genes such as FOXP2 are not sufficient as either quality control or cluster annotation.

*We thank the author for this suggestion. We have now included this figure to demonstrate the quality of our cell type clusters in **Figure S2**. We agree with the author that individual genes are not a sufficient method for cluster annotation, which is why we use an annotated reference dataset for label transfer on our data that utilizes the whole expression profile of each cell (Methods). We use the cluster-wise gene expression plots to confirm the automatic label transfer is also consistent with marker genes as other single cell studies of the dorsal striatum in rodents.*

Reviewer 2:

Overall, the authors have addressed almost all of my concerns from prior review (at a different Nature title). The revised manuscript contains a number of significant text changes and additional analyses (e.g., Fig. S1, S2, S3, and S6). The results are noteworthy and report a very useful resource for the field. While I am supportive of the manuscript in its current written form (and based on the response to reviewers and the main text), it appears that upload/transfer issues may have resulted in the lack of revised main figures with this document. A couple key issues are outlined below:

We are happy that most of the reviewer's previous feedback was addressed. They have each contributed to re-shaping our manuscript. We thank the reviewer for their support for our manuscript as a resource for the field. We sincerely apologize that the uploaded files were not appropriately transferred between journals. We have communicated this unintended problem point to the editorial staff to resolve the confusion to ease the review process.

1. The authors state in the response to reviewers that they have removed the original Figure 6, and the new manuscript text includes a figure caption for a new Figure 6 with different content. However, the old Figure 6 (containing a summary illustration of all findings, titled "Parallel cellular and molecular pathways in neurodegeneration and opioid use disorder") is still included in the paper.

Again, we apologize for the mix up of old and new files. We believe the editorial staff will have updated these files moving forward. To ensure the availability of the main figures the reviewer, we have both updated files both uploaded as separate pdf files as well as embedded them in the main text and supplements. We hope these redundant measures allow appropriate communication of our updated figures to the reviewer.

2. Similarly, despite noted changes in the response to reviewers, I could not find any changes to the main figures since the original submission - perhaps the original figures were accidentally transferred over?

See above.

3. The review files appear to contain two versions of the supplemental materials - one is updated with new content and another is the original version (with no changes).

See above.

4. The response to reviewers notes that comparison of the relative proportions of all cell types across biosamples is now included in Figure 1, but I could not find this information - again suggesting that new main figures were not included.

See above.

5. The response to reviewers notes that PMID 30257220 has been included in the reference list, however I could not find this citation in the manuscript.

We thank the reviewer for pointing out this lapse. This reference is now updated within the text at relevant points of the manuscript.

Overall, these issues make it difficult to fully evaluate the manuscript in its present form, although I think - from reading the response to reviewers - that it is likely this simply resulted from an upload/transfer error. Nevertheless, this issue should be corrected before I would be able to evaluate the full scope of the revisions.

We hope our edits will allow for a complete review of our updates to the manuscript after the transfer.

REVIEWERS' COMMENTS

Reviewer #1 (Remarks to the Author):

I have no further comments.

Reviewer #2 (Remarks to the Author):

The authors have addressed my previous comments. I am fully supportive of publication of this useful resource and striking analyses in Nature Communications.